# Nuclear-embedded mitochondrial DNA sequences in 66,083 human genomes

Wei Wei[1,2], Katherine R. Schon[1,2,3], Greg Elgar[4], Andrea Orioli[4], Melanie Tanguy[4], Adam Giess[4], Marc Tischkowitz[3], Mark J. Caulfield[5] & Patrick F. Chinnery[1,2 ✉]

DNA transfer from cytoplasmic organelles to the cell nucleus is a legacy of the endosymbiotic event—the majority of nuclear-mitochondrial segments (NUMTs) are thought to be ancient, preceding human speciation[1–3]. Here we analyse whole-genome sequences from 66,083 people—including 12,509 people with cancer—and demonstrate the ongoing transfer of mitochondrial DNA into the nucleus, contributing to a complex NUMT landscape. More than 99% of individuals had at least one of 1,637 different NUMTs, with 1 in 8 individuals having an ultra-rare NUMT that is present in less than 0.1% of the population. More than 90% of the extant NUMTs that we evaluated inserted into the nuclear genome after humans diverged from apes. Once embedded, the sequences were no longer under the evolutionary constraint seen within the mitochondrion, and NUMT-specific mutations had a different mutational signature to mitochondrial DNA. De novo NUMTs were observed in the germline once in every $10^4$ births and once in every $10^3$ cancers. NUMTs preferentially involved non-coding mitochondrial DNA, linking transcription and replication to their origin, with nuclear insertion involving multiple mechanisms including double-strand break repair associated with PR domain zinc-finger protein 9 (PRDM9) binding. The frequency of tumour-specific NUMTs differed between cancers, including a probably causal insertion in a myxoid liposarcoma. We found evidence of selection against NUMTs on the basis of size and genomic location, shaping a highly heterogenous and dynamic human NUMT landscape.

The transfer of genes from cytoplasmic organelles to the cell nucleus underpins the endosymbiotic theory of the origin of mitochondria[3]. Higher-order organisms have progressively smaller mitochondrial genomes, reflecting the translocation of mitochondrial genes into the nuclear genome over evolutionary time, facilitating the co-ordinated synthesis of organellar proteins by the cytosolic translational machinery[2]. This process has left fragments of non-expressed mitochondrial DNA (mtDNA) throughout the non-coding space, with many NUMTs being shared across species, reflecting their ancient origin[4]. Recently, whole-genome sequencing (WGS) has identified ultra-rare NUMTs in humans[5], implying that mtDNA–nuclear transfer is an ongoing process, but the rate of germline NUMT formation remains unknown. Novel mtDNA-nuclear incursions have important implications, as they can potentially disrupt protein-coding genes, causing disease[6–9], and create artefacts resembling mixed populations of mtDNA[10,11] (pseudo-heteroplasmy). Inadvertently interpreting the NUMT sequence as a mtDNA variant could confound the diagnosis of mitochondrial diseases[12] and raise questions about the possible paternal inheritance of mtDNA[13].

Large-scale WGS projects present an opportunity to characterize human NUMTs in greater depth than in other species. Here we describe the landscape of human NUMTs in 66,083 individuals, including 8,201 mother–father–child trios and 12,509 tumour–normal tissue pairs within the 100,000 Genomes Project in England. This provides a resource for the interpretation of mtDNA variants across diverse populations and for our understanding nuclear genome evolution. The results are available in a searchable online database as https://wwei.shinyapps.io/numts/.

## Atlas of human germline NUMTs

We initially studied 68,348 genomes from 67,875 participants in the Genomics England Rare Disease Project[14]. After all quality control (QC) steps (Methods), we studied 25,436 males and 28,138 females from 0 to 99 years of age (Extended Data Fig. 1a,b), including 8,201 trios whose reported relatedness was consistent with genomic predictions (Methods). Using a validated short-read NUMT detection pipeline[5,15] (Fig. 1a), we identified 335,891 NUMTs that are not present in the reference sequence based on at least two discordant read pairs detected in 53,535 individuals (>99.9%), including 3,829 different NUMTs (Extended Data Fig. 1d,e). Increasing the stringency for NUMT detection to at least 5 discordant read pairs refined the yield to 254,195 NUMTs (1,637 distinct NUMTs in 53,507 (99.87%) individuals) that are not present in the reference sequence (Fig. 1b,c and Supplementary

[1]Department of Clinical Neuroscience, School of Clinical Medicine, University of Cambridge, Cambridge, UK. [2]Medical Research Council Mitochondrial Biology Unit, University of Cambridge, Cambridge, UK. [3]Academic Department of Medical Genetics, School of Clinical Medicine, University of Cambridge, Cambridge, UK. [4]Genomics England, London, UK. [5]William Harvey Research Institute, Queen Mary University of London, London, UK. ✉e-mail: pfc25@cam.ac.uk

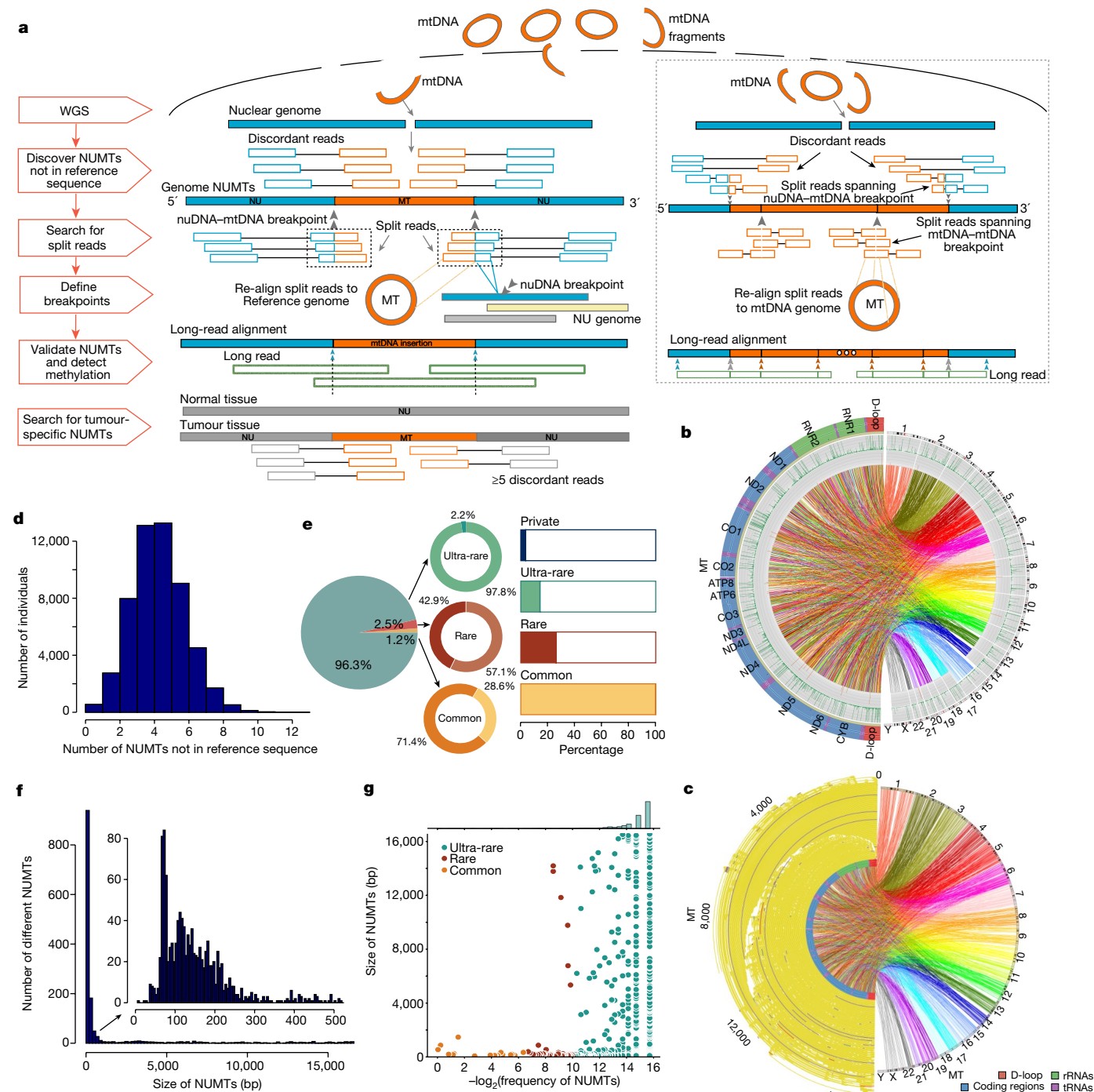

**Fig. 1 | NUMT detection in 53,574 individuals. a**, Bioinformatics pipeline for detecting NUMTs that are not present in the reference sequence, including concatenated NUMTs (boxed). Short reads: mtDNA is shown in orange, nuclear DNA (nuDNA) is shown in blue. Long reads are shown in green. MT, mitochondrial genome; NU genome, nuclear genome. **b**, 1,637 distinct NUMTs were detected in 53,574 individuals. From the outside: (1) nuclear chromosomes (right) and mtDNA genes (left); (2) frequencies of ultra-rare and rare NUMTs; (3) frequencies of common NUMTs; (4) links connect the mtDNA and nuclear breakpoints. **c**, mtDNA fragments of the 1,637 distinct NUMTs from 53,574 individuals. Left, size and location of NUMTs on mtDNA. Links connect mtDNA fragments and nuclear insertion site. **d**, The average number of NUMTs per

individual that is not present in the reference sequence and was detected by at least five discordant reads. **e**, Left, the proportion of NUMTs by population frequency (common, $F \geq 1\%$; rare, $0.1\% \leq F < 1\%$; and ultra-rare, $F < 0.1\%$). Middle, donut plots show the proportion of known (darker colour) and newly (lighter colour) identified NUMTs. Right, bar charts show the frequency of individuals carrying common, rare, ultra-rare and private NUMTs. 99.87% of individuals carry at least one common NUMT ($F > 1\%$), 26.2% of individuals carry at least one NUMT with $F < 1\%$, 14.2% of individuals carry at least one NUMT with $F < 0.1\%$ and 3.6% of individuals carry at least one private NUMT. **f**, Size distribution of germline NUMTs. NUMTs smaller than 500 bp are shown in the inset. **g**, Correlation between NUMT frequency and size.

Table 1). This higher-stringency dataset forms the basis of the results reported here, in which we refer to NUMTs as common (frequency ($F$) $\geq 1\%$), rare ($0.1\% \leq F < 1\%$), ultra-rare ($F < 0.1\%$) or private (detected

in only one family). Long-read sequencing validated our NUMT calling pipeline in 99% of cases (182 out of 184 NUMTs from 39 individuals; Fig. 1a) (Methods).

Individuals had an average of 4.7 NUMTs (s.d. = 1.6) that were not present in the reference sequence (Fig. 1d). There was no difference between males and females ($P$ value = 0.834, Wilcoxon rank-sum test; Extended Data Fig. 1f) or with age ($P$ value = 0.95, Pearson's correlation; Extended Data Fig. 1g). A total of 1,615 distinct NUMTs (98.7%) seen in 26.2% of individuals were not present in the reference sequence and were rare or ultra-rare ($F < 1$%), 1,567 different NUMTs (96.1%) seen in 14.2% of individuals were ultra-rare ($F < 0.1$%), and 1,039 (63.7%) NUMTs seen in 3.6% of individuals were private (NUMTs detected in only one family) (Fig. 1e and Extended Data Fig. 1d). As expected, the majority (71.4%) of the common NUMTs ($F \geq 1$%) had been reported previously[16–19] (Supplementary Table 2). Thus, combining the rare and ultra-rare NUMTs with the common NUMT data, we identified 1,564 NUMTs that, to our knowledge, had not been reported previously (Fig. 1e) (Methods). Defining mtDNA breakpoints at both ends (Fig. 1a), NUMTs ranged in size from 24 bp to the whole of the mitochondrial genome (median 156 bp, mean 1,597 bp and s.d 3,651 bp). The majority of NUMTs were short insertions (63.2% of NUMTs were less than 200 bp and 77.8% were less than 500 bp in size) (Fig. 1f), with an inverse relationship between NUMT size and the population frequency ($P = 0.021$, $R^2 = -0.058$, Pearson's correlation test; Fig. 1g), consistent with ongoing selection against large NUMTs. In keeping with this, we observed major differences in the frequency and distribution of NUMTs between different ethnic groups, with African and East Asian individuals being the most distinct in relation to the NUMT frequencies and chromosomal locations involved (Fig. 2 and Extended Data Fig. 2).

Some NUMTs exhibited complex structures, identified by detecting split reads mapping only to mtDNA followed by stringent QC filtering (Fig. 1a and Extended Data Fig. 3a–d). Analysis of 5,885 mtDNA–mtDNA split reads found in 3,197 trios showed that 544 were inherited from the fathers and 560 were from the mothers. One-hundred and eleven individuals shared the same rare mtDNA–mtDNA split reads within 58 ultra-rare NUMTs ($F < 0.1$%), as seen for likely concatenated NUMTs[5] (Extended Data Fig. 3a,b). Oxford nanopore long-read sequencing was performed on five families (Fig. 1a and Extended Data Fig. 3c), validating the concatenated NUMT structure. Oxford nanopore sequencing also enabled us to determine the methylation status[20] of NUMTs detected in 39 individuals (Fig. 3a) (Methods). The examined NUMTs showed increased CpG methylation relative to true mtDNA reads, which are not methylated[21], including paternally-transmitted concatenated 'mega-NUMTs' that share the same methylation pattern across two generations (Fig. 3b,c and Extended Data Fig. 3e), suppressing their expression within the nuclear genome[22]. Concatenated NUMTs can mimic the paternal transmission of mtDNA, generating a mixed haplotype resembling mtDNA heteroplasmy[5]. Here we show they increase the likelihood of detecting mixed alleles resembling heteroplasmy (when compared with the individuals not carrying concatenated NUMTs; $P < 6.02 \times 10^{-8}$ for allele fractions (AF) > 2%, $P < 3.09 \times 10^{-15}$ for AF > 1%; Wilcoxon rank-sum test) (Extended Data Fig. 3f).

Analysis of NUMT segregation in 8,201 complete mother–father–child trios revealed three private NUMTs from two families that were not seen in either parent, indicating a de novo germline NUMT mutation rate of $2.44 \times 10^{-4}$ per generation (95% confidence interval $2.95 \times 10^{-5}$ to $8.81 \times 10^{-4}$) (Fig. 3d and Extended Data Fig. 4). In each case, the de novo NUMT sequence did not align with any other site in assemblies of the nuclear genomes of the child, making it unlikely that the NUMTs originated from within the nuclear DNA. None of other NUMTs detected in each child and their parents carried the same NUMT sequence as the de novo NUMT insertions, even after increasing the mapping sensitivity by dropping the requirements from at least five discordant reads to two discordant reads. The de novo NUMTs were also not present in the reference genome or in published lists of NUMTs (Supplementary Table 2). The de novo NUMT frequency is likely to be an underestimate because of the difficulty of determining the origin of short NUMTs, although we cannot absolutely exclude the possibility of apparent

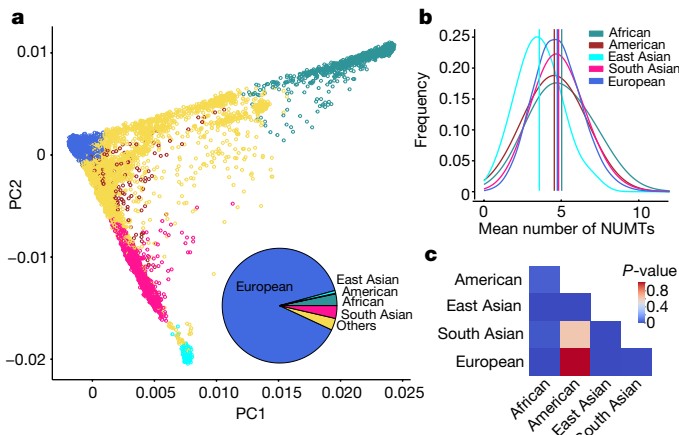

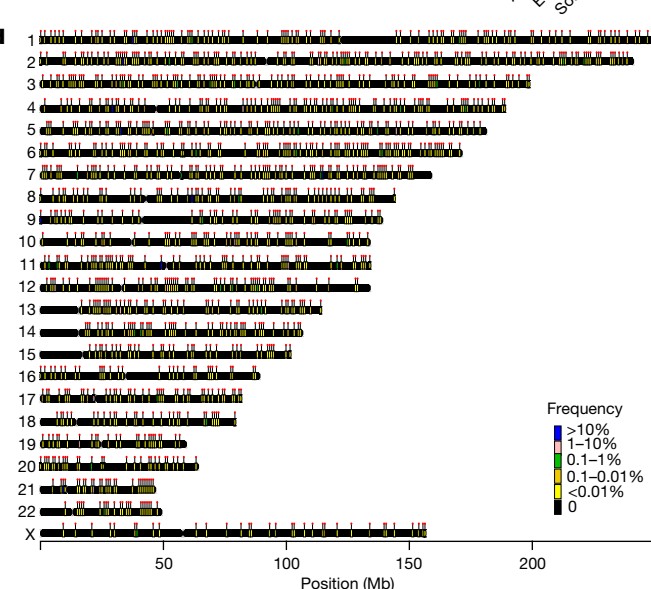

**Fig. 2 | NUMTs in the different populations. a**, Nuclear genotypes at common single nucleotide polymorphisms (SNPs) projected onto two leading principal components (PC1 and PC2). Individuals are coloured according to the assigned ancestry of their nuclear genome. The pie chart shows the proportion of each group overall: East Asian (cyan), South Asian (pink), African (green), American (red), European (blue) and unassigned (yellow). **b**, The average number of NUMTs detected in populations with different ancestries. Vertical lines show the average number of NUMTs from each population. **c**, Heat map showing $P$ values from pairwise comparison of the average number of NUMTs detected between populations of different ancestries (two-sided Wilcoxon rank-sum test). **d**, Chromosomal locations of NUMT insertions detected in this study, coloured by the frequency of NUMTs. Dots show the locations of the NUMTs. Chromosomal locations of different NUMT insertions detected for each ancestry are shown in Extended Data Fig. 2.

de novo NUMTs arising from other parts of the nuclear genome and as opposed to a new mtDNA insertion event.

## Characteristics of NUMT insertions

Next, we studied the mtDNA and nuclear DNA context of the NUMTs, which were found on all nuclear chromosomes (Fig. 2d) and involved the entire mtDNA (Fig. 1b,c). The 3,184 corresponding mtDNA breakpoints were enriched in the non-coding displacement loop (D-loop) ($P = 0.001$)—particularly in three hypervariable segment regions (HV1, $P = 0.002$; HV2, $P = 0.001$; and HV3, $P = 0.006$)—and both heavy strand (OHR, $P = 0.002$) and light strand (OLR, $P = 0.016$) origins, and were less likely to involve *MT-ATP6* ($P = 0.001$), *MT-ND2* ($P = 0.015$) and *MT-ND3* ($P = 0.034$) (Fig. 3e and Extended Data Fig. 5a,b). This was supported by

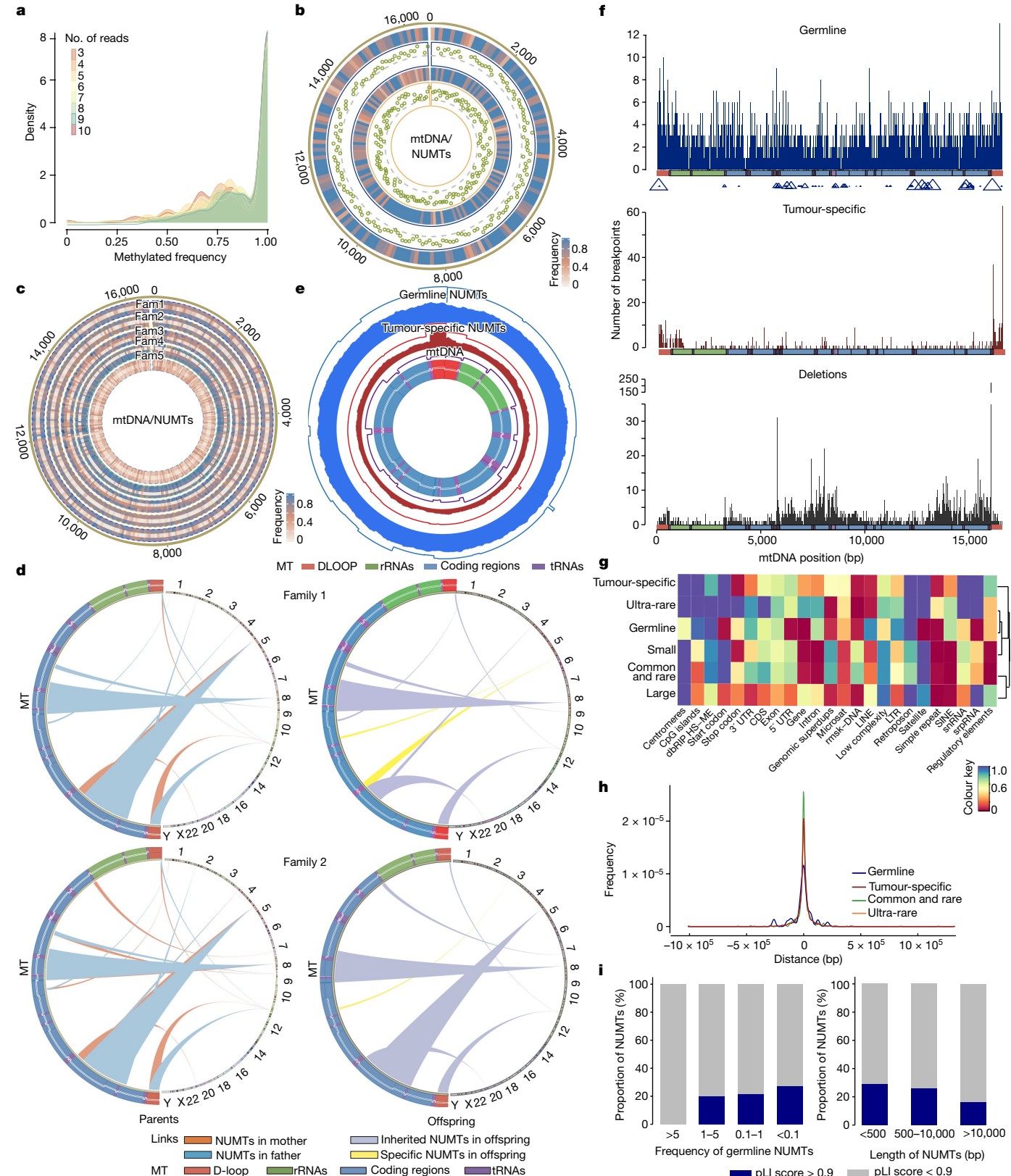

**Fig. 3** | See next page for caption.

the distribution of the mtDNA fragments (*P* = 0.03, odds ratio = 1.14, 95% confidence interval 1.01–1.28, Fisher's exact test) (Fig. 3e). There was a weak correlation between the germline NUMT mtDNA breakpoints and the location of known deletion breakpoints in mtDNA, which exhibited marginal significance (*P* = 0.047, *R*² = 0.24, Pearson correlation) (Fig. 3f

and Extended Data Fig. 5c). Overall, we observed a strong positive correlation between the length of each chromosome and the number of NUMTs detected on each chromosome after accounting for other genomic features (*P* = 1.42 × 10⁻⁶, linear regression test). However, chromosomes 3, 6 and 21 had a larger number of NUMTs per Mb than

**Fig. 3 | Characteristics of NUMTs in humans. a**, Methylation frequency of NUMTs in 39 individuals. Colours correspond to the number of long reads that are not affected by the sequencing depth. **b**, Methylation status of a concatenated NUMT from a father–proband pair. From the outside: (1) methylation frequency of the concatenated NUMT in the father; (2) the ratio of methylation frequency between the NUMT and the non-methylated mtDNA sequence in the father; (3) methylation frequency of the concatenated NUMT in the proband; (4) the ratio of methylation frequency between the NUMT and the non-methylated mtDNA sequence in the proband. Green dots show methylated sites. This analysis includes only reads that were definitively nuclear in origin. The colour corresponds to the methylation frequency. **c**, Methylation profile for five families (fam1–fam5) with concatenated NUMTs (Supplementary Table 7). From the outside: father, mother, sibling (when available) and proband. Individuals harbouring concatenated NUMTs had higher methylation levels than the individuals without concatenated NUMTs. The colour corresponds to the methylation frequency. **d**, Three de novo NUMTs from two trios.

**e**, The frequency of mtDNA insertion from germline and tumour-specific NUMTs. From the outside: (1) frequencies of breakpoints from germline NUMTs; (2) frequencies of mtDNA fragments from germline NUMTs; (3) frequencies of breakpoints from tumour-specific NUMTs; (4) frequencies of mtDNA fragments from tumour-specific NUMTs; (5) frequencies of mtDNA sequences expected by chance; (6) mtDNA regions. **f**, Distribution of breakpoints on mitochondrial genes with germline NUMTs, tumour-specific NUMTs and mitochondrial deletions (window size = 100 bp). The triangle size indicates the frequency of NUMTs within each window. **g**, P values for enrichment analysis of different genome regions (Supplementary Figs. 1–3 and Methods). Microsat, microsatellite; rmsk-DNA, repetitive DNA; snRNA, small nuclear RNA; srpRNA, signal recognition particle RNA; superdups, superduplications. **h**, The distance of NUMT locations from the TSS. **i**, The proportion of NUMTs within genes with high and low pLI scores grouped by NUMT frequency (left) and grouped by NUMT size (right).

the remaining autosomes (chromosome 3, $P = 0.03$; chromosome 6, $P = 0.005$; chromosome 21, $P = 0.03$, two-tailed permutation test), and the X chromosome had a reduced number of NUMTs per Mb ($P = 0.001$). Two hundred and twenty-eight NUMTs were observed on the X chromosome, with the expected approximately twofold higher number in females than males (151 of the 28,138 females, and 75 of the 25,426 males; Fisher exact test $P = 1.713 \times 10^{-5}$, odds ratio = 1.824, 95% confidence interval 1.374–2.441). The Y chromosome was not analysed owing to the complex duplicated structure limiting confident alignment.

Previous reports of local sequence characteristics associated with NUMT insertion[23] prompted a comprehensive analysis of the proximity of the unique NUMTs to the centromere, genomic duplications, simple repeats, dbRIP HS-ME (retrotransposon insertion polymorphisms, human-specific mobile elements), regulatory elements, CpG islands, satellites and retrotransposons (including long interspersed elements (LINEs) and short interspersed elements (SINEs)). Common and rare NUMTs ($F \geq 0.1\%$) were more likely to occur near or within genomic duplications ($P = 0.030$), and ultra-rare NUMTs were enriched in regulatory elements ($P = 0.011$), SINEs ($P = 0.003$), simple repeats ($P = 0.006$) and introns $P = 0.003$ (Fig. 3g and Supplementary Figs. 1–3). No common NUMTs were within the 500 bp region flanking transcription start sites (TSS), consistent with selection against NUMTs that disrupt gene function (Fig. 3h and Extended Data Fig. 5d). Consistently, gene tolerance scores[24] (pLI) were inversely correlated with the frequency of NUMTs in the population (Fig. 3i).

## Atlas of tumour-specific NUMTs

Next, we studied 26,488 cancer WGS from the Genomics England Cancer project. After QC steps (Methods), we analysed 12,509 paired WGS from tumours and healthy tissues representing the germline for 21 cancer types (Extended Data Fig. 6a–d and Supplementary Table 3). Overall, tumours had a higher mean number of NUMTs ($6.5 \pm 2.2$ (mean ± s.d.)) that were not present in the reference sequence than the corresponding normal tissue ($4.8 \pm 1.6$; $P < 2.2 \times 10^{-16}$, Wilcoxon rank-sum test) (Fig. 4a and Supplementary Fig. 4). This difference probably reflects the tumour itself, rather than the normal tissue in each case, because the mean number of NUMTs did not differ between different normal tissue types (the average detected NUMT was 4.7 in saliva cells, 5 in skin fibroblasts and 4.9 in blood samples; saliva versus blood, $P = 0.24$, estimate = −0.1; fibroblast versus blood $P = 0.67$, estimate = −0.1, linear regression test) (Extended Data Fig. 6e). The frequency of cancer germline NUMTs was not different from the frequency of germline NUMTs measured in the Rare Disease Project participants ($P = 0.924$, linear regression test accounting for sequencing depth) (Extended Data Fig. 6f). There were no sex differences in the NUMT distribution (Supplementary Fig. 5). For most tumours, there was no correlation between the age of an individual at diagnosis and the number of NUMTs

(Extended Data Fig. 6b and Supplementary Fig. 6a). However, the mean number of NUMTs was lower in haematological malignancies from older individuals, probably reflecting their origin in clonal haematopoiesis[25] ($P = 3.29 \times 10^{-3}$, estimate = −0.007, linear regression).

Next, we focussed on a subgroup of the tumour-specific NUMTs that were not present in any other non-cancer genome, which provided high confidence that these NUMTs arose either in somatic tissues leading to the cancer, or in the cancer itself. Three hundred and seventy nine of these de novo NUMTs were seen in 251 tumours (2.3%) from 10,713 tumour–normal pairs, giving a rate of $3.56 \times 10^{-2}$ per cancer per genome (95% confidence interval $3.38 \times 10^{-2}$ to $3.74 \times 10^{-2}$) (Fig. 4b,c and Supplementary Table 4; Methods), which is higher than the germline rate ($P = 2.08 \times 10^{-59}$, Fisher's exact test) and consistent with previous reports[15,26]. Eighty-two tumours carried more than one de novo NUMT, which was more than expected by chance ($P < 2.2 \times 10^{-16}$, Fisher's exact test) (Fig. 4b). The mean number of tumour-specific NUMTs was 0.035 (s.d. = 0.29), with a median length of 396 bp (first quartile 250 bp, third quartile 524 bp, mean = 1,197 bp), which was higher than the number of germline NUMTs (median = 156 bp, first quartile 97 bp, third quartile 382 bp) ($P < 2.2 \times 10^{-16}$, Wilcoxon rank-sum test) (Fig. 4d,e). These findings are consistent with cancer driving NUMT formation. The proportion of tumours with a de novo NUMT depended on the tumour type, with renal and colorectal tumours having fourfold fewer NUMTs than breast cancers ($P = 1.93 \times 10^{-6}$, Fisher's exact test) and around 7.5-fold fewer than bladder cancers ($P = 3.42 \times 10^{-4}$, Fisher's exact test) (Fig. 4f), which had more NUMTs than the other tumour types (Fig. 4g,h), as shown previously[26]. The average number of tumour-specific NUMTs did not correlate with age (Supplementary Fig. 6b), implying that they arose during carcinogenesis, and not in somatic cells throughout life before cancer formation. A comparison with ref. [26] is shown in Supplementary Table 5.

## The signature of de novo NUMTs in cancer

The mtDNA segments forming de novo tumour NUMTs differed from those in the germline (Fig. 3e): they were less likely to involve *MT-CO3* ($P = 7.7 \times 10^{-3}$), *MT-ND4* ($P = 3.1 \times 10^{-3}$), *MT-ND4L* ($P = 3.4 \times 10^{-3}$) and *MT-ND5* ($P = 5.3 \times 10^{-3}$), but more than 2.5-fold more likely to involve the D-loop ($P = 3.36 \times 10^{-36}$), largely because of an approximately four-fold over-representation of breakpoints in termination-associated sequence 2 (TAS2) ($P = 1.03 \times 10^{-7}$, Fisher's exact test) (Extended Data Fig. 5a,b), also reflected in the mtDNA fragments (D-loop, $P = 5.51 \times 10^{-30}$, odds ratio = 2.00, 95% confidence interval 1.77–2.25, Fisher's exact test) (Figs. 3e and 4c). This could explain the observed correlation between de novo NUMT breakpoints and known mtDNA deletion breakpoints ($P = 0.004$, $R^2 = 0.44$, Pearson correlation) (Fig. 3f and Extended Data Fig. 5c), which also tend to cluster around the D-loop at the 3' end[27]. Tumour-specific NUMTs were more common on

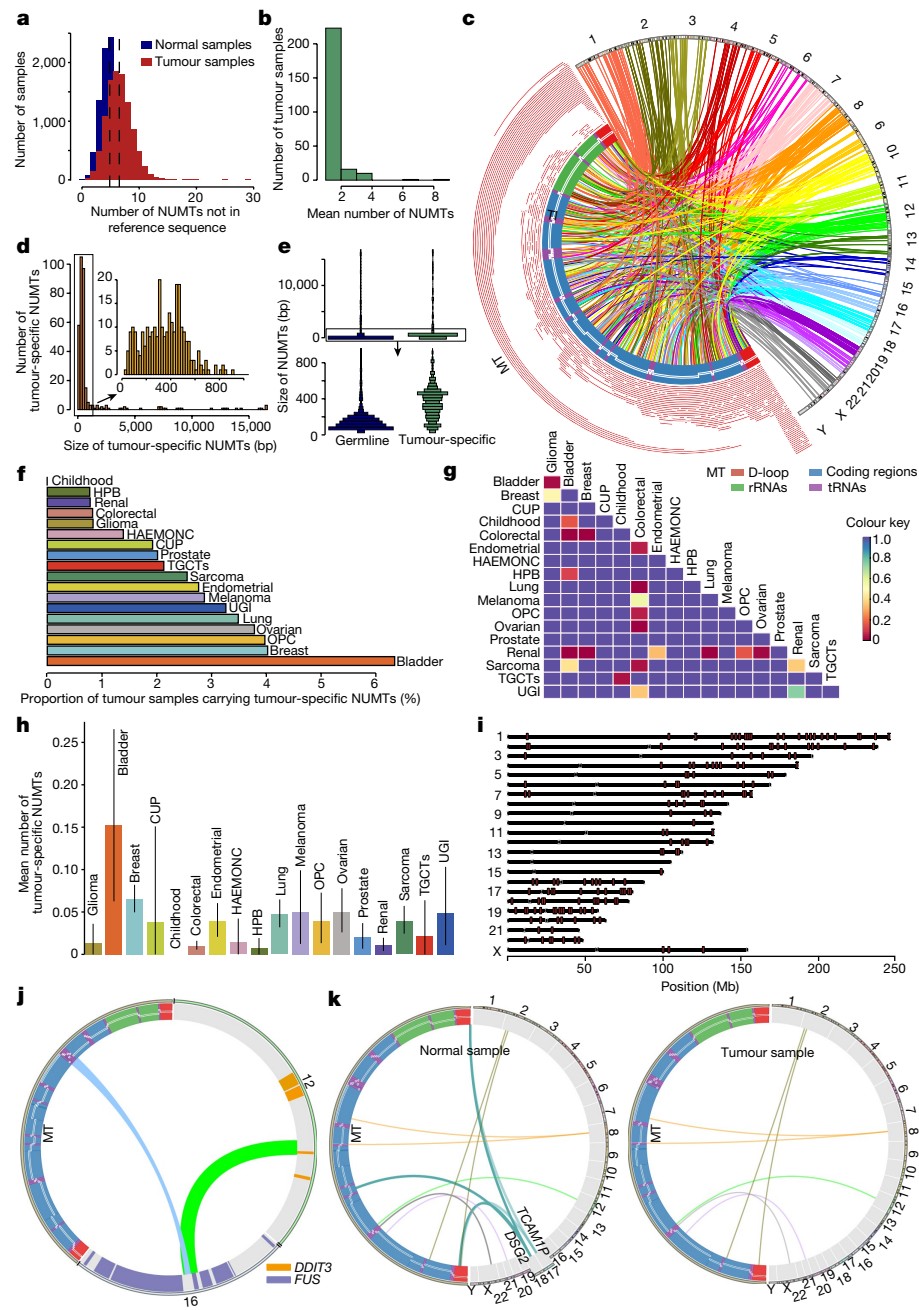

**Fig. 4 | NUMTs in human cancers. a**, Average number of NUMTs detected per normal and tumour sample that are not present in the reference sequence. **b**, Average number of tumour-specific NUMTs detected in tumours. **c**, Tumour-specific NUMTs detected in 12,509 normal–tumour pairs. Left, NUMT size and location on mtDNA. Links connect breakpoints between mtDNA and nuclear genomes. **d**, Size distribution of tumour-specific NUMTs (red) and tumour-specific NUMTs smaller than 1,000 bp (orange). **e**, Size distribution of all germline and tumour-specific NUMTs (top) and germline and tumour-specific NUMTs smaller than 1,000 bp (bottom). **f**, The percentage of different types of tumours with at least one tumour-specific NUMT. **g**, $P$ values from pairwise comparison of the average number of tumour-specific NUMTs from different tumour types. **h**, Average number of tumour-specific NUMTs for each tumour type. Data are mean ± s.e.m. Glioma, $n = 359$; bladder, $n = 268$; breast, $n = 2,038$; CUP, $n = 52$; childhood, $n = 170$; colorectal, $n = 1,934$; endometrial, $n = 579$;

HAEMONC, $n = 72$; HPB, $n = 258$; lung, $n = 1,061$; melanoma, $n = 244$; OPC, $n = 151$; ovarian, $n = 423$; prostate, $n = 298$; renal, $n = 1,022$; sarcoma, $n = 979$; TGCTs, $n = 47$; UGI, $n = 184$. **i**, Chromosomal locations of tumour-specific NUMTs, shown as red bars. **j**, NUMTs involved in *FUS–DDIT3* chimeric fusion. NUMTs are shown as a blue link and the *FUS–DDIT3* fusion is shown as a green link. The chromosome number and mitochondrial genome are indicated. **k**, Example of lost NUMTs in a breast tumour sample. The links represent NUMTs detected in either normal (left) or tumour (right) samples. The chromosome number and mitochondrial genome are indicated. CUP, carcinoma of unknown primary; endometrial, endometrial carcinoma; glioma, adult glioma; HAEMONC, haemato-oncology; HPB, hepato-pancreato-biliary cancer; melanoma, malignant melanoma; OPC, oral and oropharyngeal cancers; TGCTs, testicular germ cell tumours; UGI, upper gastrointestinal cancer.

chromosome 19 ($P = 9.08 \times 10^{-6}$) and less common on chromosome 6 ($P = 1.53 \times 10^{-3}$) (Fig. 4i) and were more likely to involve repetitive elements ($P = 4.24 \times 10^{-16}$), particularly satellite repeats ($P = 0.023$)

and microsatellites repeats ($P = 0.007$) than the germline NUMTs (Fig. 3g and Supplementary Fig. 1). Finally, a greater proportion of tumour-specific NUMTs were found within 500 bp, 2,000 bp and

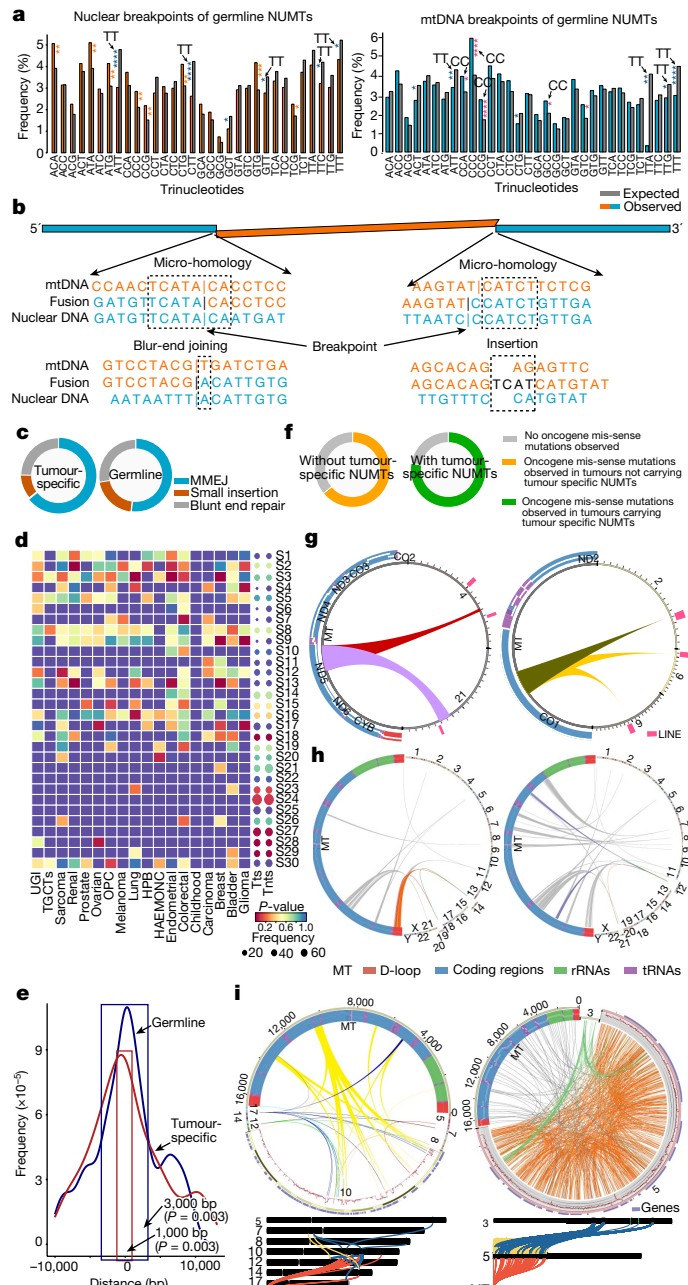

**Fig. 5 | Molecular mechanism of NUMT formation. a**, Trinucleotide frequencies around NUMT breakpoints in the nuclear genome (left) and mtDNA (right) (details in Extended Data Fig. 8a). Arrows point to the nCC/CCn or nTT/TTn trinucleotides significantly enriched in NUMTs. *$P < 0.05$, **$P < 0.01$, ***$P < 0.001$, ****$P < 0.0001$. **b**, Microhomology-mediated end joining during formation of NUMTs. **c**, The proportion of microhomology sequences, small insertions and blunt-end joining between nuclear and mtDNA sequences around NUMT breakpoints. **d**, Cancer signature enrichment for each cancer type (heat map) and all cancer types (dots). Dot size is proportional to the number of samples with each signature in tumour-specific NUMTs (Tts) and non-tumour-specific NUMTs (Tnts). **e**, The distance between NUMTs and PRDM9-binding sites in germline and tumour-specific NUMTs. **f**, NUMTs in tumours with and without missense mutations in human DNA repair genes. **g**, Two examples of the same mtDNA fragment detected at two locations in the nuclear genome, showing evidence that the NUMT inserted into one location and then moved to another. **h**, Left, an mtDNA fragment inserted into chromosome 14 and 19, and a translocation between chromosome 14 and 19. NUMTs were detected on chromosomes 14 and 19, suggesting that the NUMTs inserted into the nuclear genome before translocation occurred, then moved to another location with the translocation. Right, an mtDNA fragment inserted into chromosome 12, and a translocation between chromosome 12 and 21. NUMTs were seen on chromosome 12, but not on chromosome 21, suggesting that the NUMTs inserted into the nuclear genome after translocation occurred. **i**, Two examples of samples carrying mito-chromothripsis observed in this study. Circos plots show the locations of NUMTs in both nuclear and mtDNA genomes, and the structural variants in the nuclear genome. Nuclear genome sequencing depth is shown in the red line. Chromosome maps show the structural variants involved in multiple chromosomes in the nuclear genome. The read alignment from Integrated Genomics Viewer is shown in Extended Data Fig. 9c,d.

inserted into genes on the COSMIC Cancer Gene Census list[28] (two in *FHIT*, which is a fragile genomic site[29], and one each in *CTNNA2*, *DDIT3*, *WIF1*, *BCL11B*, *KDM5A* and *AKT2*) (Supplementary Table 4). One tumour had a NUMT insertion in an intron of *FANCI*, which is involved in DNA repair. Complex rearrangements with NUMT insertion at the site of chromosomal translocations were also seen in three out of eight tumour samples (Fig. 4j and Extended Data Fig. 7a). One myxoid liposarcoma tumour carried a *FUS–DDIT3* chimeric fusion oncoprotein caused by a complex rearrangement involving a NUMT insertion (Fig. 4j and Extended Data Fig. 7a). FUS–DDIT3 fusions are present in 90% of myxoid liposarcomas[30], implicating NUMT in carcinogenesis in the individual in our study. Three private NUMTs in non-tumour tissue were not found in the matched breast tumours, potentially influencing prognosis through the loss of *DSG2*[31] and *TCAM1P*[32] (Fig. 4k and Extended Data Fig. 7b). Two normal tissues from individuals with haematological cancer carried extremely high numbers of NUMTs that were not present in the tumour tissues (Extended Data Fig. 7c), probably reflecting clonal proliferation.

## NUMT insertion and modification

NUMT breakpoints were more likely to involve nCC/CCn trinucleotides on the mtDNA genome and less likely to involved nTT/TTn on both the nuclear genome and mtDNA (Fig. 5a, Extended Data Fig. 8a and Supplementary Table 6). Extending the analysis to 2 bp, 3 bp and 4 bp beyond the mtDNA breakpoint showed that poly-C tracts were 8, 12 and 18 times more numerous than expected by chance ($P = 7.57 \times 10^{-10}$, $P = 2.13 \times 10^{-5}$ and $P = 6.3 \times 10^{-5}$), implicating microhomology in NUMT insertion events through recombination. We also observed overlapping sequence microhomology ($\geq 1$ bp) in 51.9% of the NUMT breakpoints ($P = 2.05 \times 10^{-45}$, Fisher's exact test), consistent with microhomology-mediated end joining (MMEJ) during some NUMT formation; blunt-end repair in 27.6% of the NUMT breakpoints and short-nucleotide insertions in 20.5% of the NUMT breakpoints, implicating non-homologous end joining[33,34] (Fig. 5b,c). A greater proportion of tumour-specific NUMTs (64.1%) had overlapping sequence

5,000 bp of the TSS than for germline NUMTs (Fig. 3h and Extended Data Fig. 5d). Together, these findings suggest that a combination of local sequence characteristics, genome instability and less opportunity for selection to remove specific NUMTs due to relaxed evolutionary constraints explains why the NUMT landscape differs from the germline.

## Adverse consequences of NUMT insertion

Nine hundred and forty six (58%) germline NUMTs were observed in gene regions, with the majority (85.8%, $n = 812$) being enriched in introns versus exons ($P = 0.01$, permutation test) (Fig. 3g and Supplementary Figs. 1–3). No common or rare NUMTs ($F > 0.1\%$) were found in the coding DNA sequences (CDS) ($P = 0.039$ permutation test), and none were predicted to cause rare disease (Methods and Supplementary Information, 'Results'), consistent with NUMTs being under evolutionary constraint. Two hundred and twenty tumour-specific NUMTs were found in gene regions, including 13 in CDS, 3 affecting stop codons, 4 affecting start codons, 16 in 3' or 5' untranslated regions (UTRs). Eight tumours harboured tumour-specific NUMTs

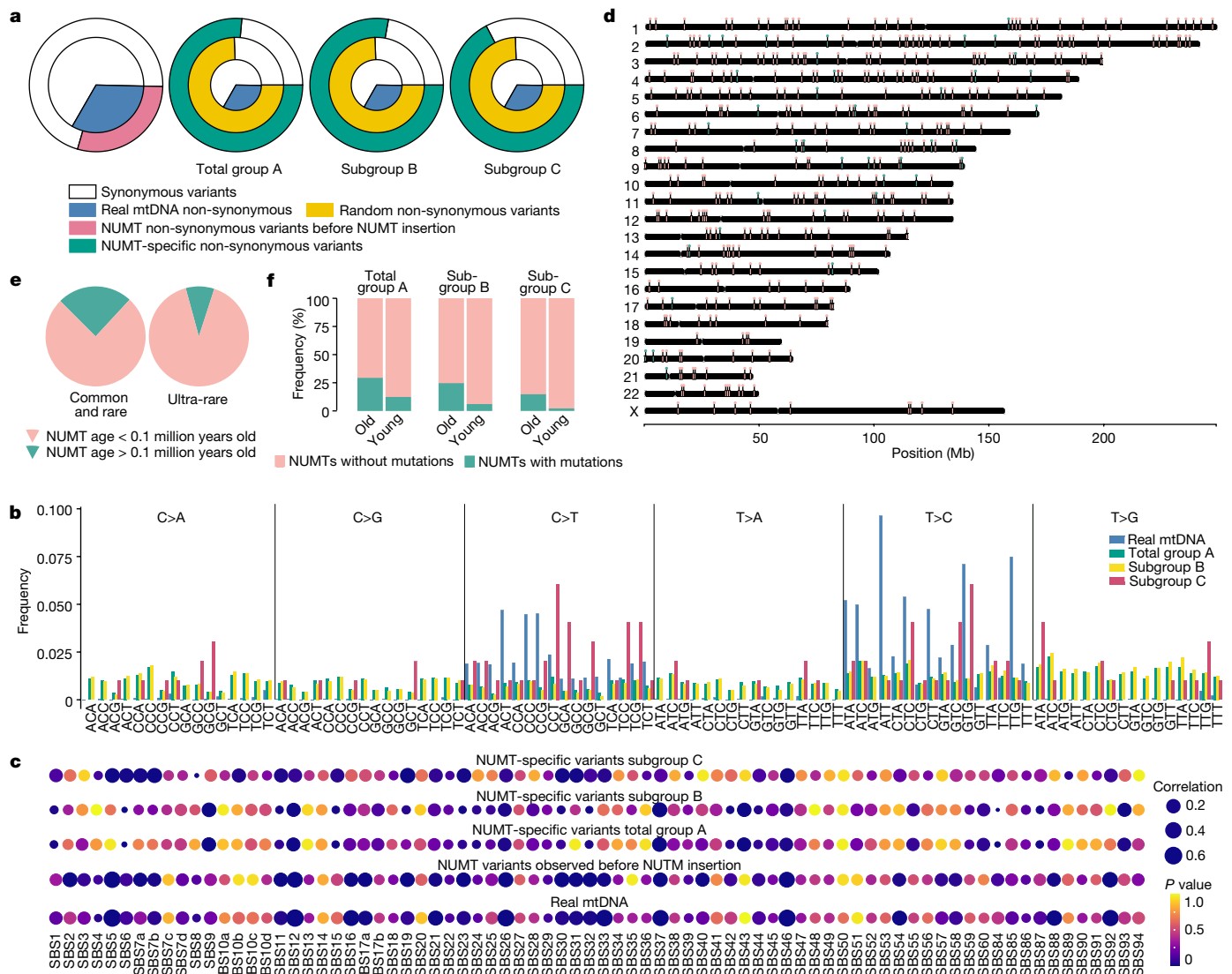

**Fig. 6 | Molecular evolution of NUMT sequences. a**, Synonymous and non-synonymous variants. The proportion of non-synonymous variants from different variant groups are shown as different colours. **b**, Trinucleotide mutational signatures. **c**, Correlation of trinucleotide mutational signatures of NUMT variants with cancer signatures. **d**, Chromosome map of NUMTs estimated to be less than 0.1 million years old (red) and those estimated to be more than 0.1 million years old (blue). **e**, The proportion of older and younger NUMTs among common and rare, and ultra-rare NUMTs. **f**, The frequency of NUMTs observed with at least one variant in older and younger NUMTs, and in total group A, subgroup B and subgroup C NUMTs.

microhomology than germline NUMTs ($P = 5.22 \times 10^{-10}$, Fisher's exact test) (Fig. 5c).

We also observed enrichment of the cancer trinucleotide mutation signatures[35] S2 ($P = 6.93 \times 10^{-7}$), S3 ($P = 4.68 \times 10^{-13}$) and S13 ($P = 1.72 \times 10^{-18}$) in cancers carrying the tumour-specific NUMTs (Fig. 5d). NUMT insertion resembles transposon jumping previously associated with S2 and S13[35], and S3 is linked to failed double-strand break (DSB) repair by homologous recombination, in which NUMTs have a role[35]. Signatures 2 and 13 are also enriched for APOBEC-mediated point mutations, which can also induce DSBs[36]. Thus there appear to be common molecular mechanisms behind somatic mutation in cancers and NUMT formation.

In keeping with this, germline NUMTs were more likely to be found within 3 kb of a PRDM9-binding site ($P = 0.003$, permutation test), and tumour-specific NUMTs were more likely to be found within 1 kb of a PRDM9-binding site ($P = 0.003$, permutation test) (Fig. 5e and Extended Data Fig. 8c). PRDM9 is implicated in DSB repair and determines crossover hotspots during meiosis[37], so co-location is consistent with NUMTs having a role in DSB repair[33,34]. Thus, several different molecular mechanisms are involved in NUMT formation, all of which are related to nuclear genome instability. In keeping with this, tumour samples carrying missense mutations in DNA repair oncogenes[38,39] were more likely to harbour tumour-specific NUMTs than the remaining tumours (77.7% versus 63.1%, Fisher's exact test $P = 5.05 \times 10^{-6}$, 95% confidence interval 1.44–2.68, odds ratio = 1.95) (Fig. 5f and Supplementary Fig. 7).

It has been suggested that NUMTs are mobile after the initial insertion event[40], hitch-hiking on other transmissible elements. We found several examples supporting this hypothesis (Fig. 5g). We also found NUMTs associated with large deletions, insertions, copy number gain or loss, and particularly at the breakpoints of complex structural variants (Extended Data Fig. 9a,b). Several cancers contained extensive NUMT rearrangements, with evidence of insertion into the nuclear genome before a nuclear–nuclear translocation (Fig. 5h). We also found two examples in which multiple fragments of the mtDNA were embedded throughout the genome across multiple chromosomes (Fig. 5i and Extended Data Fig. 9c,d) resembling the extreme rearrangements seen in chromothripsis[41] (mito-chromothripsis).

## Molecular evolution of NUMT sequences

To understand the molecular evolution of the mtDNA sequences after their insertion into the nuclear genome, we determined the complete nucleotide sequence of 931 different NUMTs incorporating 144,805 bp, where complete local assembly of NUMTs was possible from short-read sequencing (Methods). The results of this analysis are reported in Fig. 6 and Supplementary Information, 'Results'. Finally, we estimated the age of 429 NUMT insertions (Methods). The majority (more than 90%) were less than 0.1 million years old and 41 (9.5%) were more than 0.1 million years old, with a range of up to 3.75 million years (Fig. 6d and Supplementary Table 1). As expected, the older NUMTs were more common in the population (Fig. 6e), particularly in African genomes (Extended Data Fig. 8d), and were more likely to carry NUMT-specific mutations than the younger NUMTs (total group A: $P = 7.2 \times 10^{-3}$, odds ratio = 2.92, 95% confidence interval 1.27–6.39; subgroup B: $P = 3.9 \times 10^{-4}$, odds ratio = 2.92, 95% confidence interval 1.27–6.39; subgroup C: $P = 9.0 \times 10^{-4}$, odds ratio = 8.06, 95% confidence interval 2.18–28.27, Fisher's exact test) (Fig. 6f). Together these findings indicate ongoing NUMT insertion and evolution throughout human evolution.

## Discussion

NUMTs were considered ancient remnants of previous mtDNA translocation events that were often shared between related species[42]. Here we show that NUMT formation is an ongoing process, with de novo germline events occurring approximately once in every $10^4$ births and somatic insertions occurring once in every $10^3$ cancers. This leads to high NUMT diversity within the human population, with 14.2% of individuals harbouring an ultra-rare NUMT found in less than 1 in 1,000 people. The wholesale transfer of mtDNA fragments into the nucleus genome would inevitably increase the size of the human genome[3]. However, the inverse correlation between NUMT size and the frequency of its occurrence in the population points towards a selective process counter-balancing NUMT insertion, maintaining genome size and removing NUMTs that influence gene expression. Co-location of NUMTs with PRDM9-binding sites would facilitate their removal in the germline because PRDM9 determines sites of recombination hotspots during meiosis[37]. In this way, NUMTs can act as 'temporary fixes' resembling a sticking-plaster, repairing DSBs until they are removed during meiosis. The higher burden and distribution of NUMTs in cancers probably reflects a heightened state of genome instability in the absence of selection over short time periods.

Although NUMTs can involve the entire mtDNA molecule, NUMT breakpoints were more common in the non-coding D-loop, including the origins of heavy and light strand replication. This raises the possibility that mtDNA deletions are involved in NUMT formation. However, a more compelling explanation involves mtDNA transcription and associated replication, which originates in the D-loop[43]. The recent description of mitochondrial herniations and BSX–BAK macropores provide one route[44], potentially involving RNA intermediates leaking into the cytoplasm following mtDNA DSBs[45]. This could also occur in single cells, contributing to somatic mosaicism.

The translocation of organelle genes into the nucleus has had a key role in establishing the symbiotic relationship between mitochondria and eukaryotic cells. Here we show that the mechanisms of DNA transfer remain active and modify the germline approximately once in every in 4,000 births. It is therefore conceivable that an endosymboisis that began around 1.45 billion years ago is not yet complete.

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

# Methods

## Study samples

We studied 68,348 genomes from whole-blood DNA in Genomics England Rare Disease Project and 26,488 cancer genomes from Genomics England Cancer Project. DNA was extracted and processed based on the Genomics England Sample Handling Guidelines (https://legacy.genomicsengland.co.uk/about-genomics-england/the-100000-genomes-project/information-for-gmc-staff/sample-handling-guidance/). DNA samples were received in FluidX tubes (Brooks) and accessioned into Laboratory Management Information System (LIMS) at UK Biocentre. Following automated library preparation, libraries were quantified using automated quantitative PCR, clustered and sequenced. Libraries were prepared using the Illumina TruSeq DNA PCR-Free High Throughput Sample Preparation kit or the Illumina TruSeq Nano High Throughput Sample Preparation kit[46].

## Ethical approval

Ethical approval was provided by the East of England Cambridge South National Research Ethics Committee under reference number 13/EE/0325, with participants providing written informed consent for this approved study. All consenting participants in the Rare Disease arm of the 100,000 Genomes Project were enrolled via 13 centres in the National Health Service (NHS) covering all NHS patients in England.

## Quality control checks of rare disease genomes

All the samples were passed an initial QC check based on sequencing quality and coverage from the sequencing provider (Illumina) and Genomics England internal QC checks (https://research-help.genomicsengland.co.uk/display/GERE/Sample+QC). We only included the samples aligned to the *Homo sapiens* NCBI GRCh38 assembly with decoys ($N$ = 58,335). All the samples were sequenced to produce at least 85 Gb of sequence data with sequencing quality of at least 30. Alignments covered at least 95% of the genome at 15x or above with well-mapped reads (mapping quality > 10) after discarding duplicates. Additionally, all included samples have passed a set of basic QC metrics: (1) sample contamination (VerifyBamID freemix[47]) < 0.03, (2) ratio of single nucleotide variant (SNV) Heterozygous-to-Homozygous (Het-to-Hom) calls < 3, (3) total number of SNVs between 3.2 M–4.7 M, (4) array concordance > 90%, (5) median fragment size > 250 bp, (6) excess of chimeric reads < 5%, (7) percentage of mapper reads > 60%, and (8) percentage of AT dropout < 10%. 57,961 genomes were passed WGS QCs. We further excluded the samples with the average depth of mitochondrial genomes below 500x after re-aligned the mitochondrial reads (see details below). For the rare disease genomes study, we included 53,574 individuals, 25,436 male and 28,138 females, age from 0 to 99 years (Extended Data Fig. 1a,b). The average depth of WGS was 42x (s.d. = 7.7x) and average depth of mtDNA was 1,990x (s.d = 866x) (Extended Data Fig. 1c).

## Family QC checks

In the family related analysis, WGS family selection quality checks are processed for rare disease genomes, reporting abnormalities of sex chromosomes and reported versus genetic sex summary checks (computed from family relatedness, mendelian inconsistencies, and sex chromosome checks). For the sex determination, the coverage data for the X and Y chromosomes was compared to the average coverage for the sample autosomes using PLINK v1.90[48] (www.cog-genomics.org/plink/1.9/). The resulting output is compared with the participant sex provided at sample collection. Relatedness checks were based on verification of the mendelian inconsistencies between members of a trio/family. The individual VCF files were merged into a family VCF with BCFTools (v1.3.1)[49] and the mendelian inconsistencies again checked with PLINK. The relationships are also checked by calculated genomic identity-by-descent values for all pairwise relationships in a family using PLINK and comparing with expected values for reported relationship (https://research-help.genomicsengland.co.uk/). We further processed an independent relatedness check using our previously published method[50]. In brief, a list of 32,665 autosomal SNPs was selected to estimate relatedness. By filtering the merged VCF and the 1000 Genomes reference set[51] with the selected SNPs, the pc-relate function from the GENESIS package was applied to obtain the pairwise relatedness[52]. The first 20 principal components were used to weight the population structure, and the reference set was used to increase genetic diversity accounted for by the principal component analysis. Finally, we included 8,201 families whose relatedness was consistent between two independent prediction methods and the clinical records.

## QC checks of cancer genomes

We initially studied 26,488 cancer genomes from Genomics England Cancer Project. Samples were prepared using an Illumina TruSeq DNA Nano, TruSeq DNA PCR-Free or FFPE library preparation kit and then sequenced on a HiSeq X generating 150 bp paired-end reads. Germline samples were sequenced to produce at least 85 Gb of sequences with sequencing quality of at least 30. For tumour samples at least 212.5 Gb was required. Alignments for the germline sample covered at least 95% of the genome at 15x or above with well-mapped reads (mapping quality > 10) after discarding duplicates (https://research-help.genomicsengland.co.uk/).

For the sample cross-contamination checks, germline samples are processed with VerifyBamID[47] algorithm and PASS status is assigned to the samples with less than 3% of contamination. Tumour samples were processed with the ConPair algorithm[53] with a PASS status indicating contamination is below 1% as described in https://research-help.genomicsengland.co.uk/display/GERE/10.+Further+reading+and+documentation?preview=/38047056/45023724/Cancer%2520Analysis%2520Technical%2520Information%2520Document%2520v1-11%2520main.pdf#id-10.Furtherreadinganddocumentation-TechnicalDocumentation.

After the QC steps described above, 12,509 tumour–normal tissue pairs from 12,509 tumour samples and 11,913 matched normal tissue (germline) samples from 11,909 individuals remained. Samples were prepared using 5 different methods (FF, FFPF, CD128 sorted cells, EDTA and ASPIRATE) and three different library types (PCR, PCR-FFPE and PCR-free). We performed the additional QCs by comparing the average number of NUMTs were detected from the samples prepared by different methods and library types. We observed that the average number of NUMTs was significantly different between different groups (Supplementary Fig. 8a). To avoid possible bias caused by sample preparation and library type, we only included the 10,713 tumour–normal sample pairs prepared using FF and library type PCR-free from 9648 individuals across 21 cancer types (Extended Data Fig. 6a). The average WGS depth of tumour sample was 117x (s.d. 10.1x) and the average WGS depth of germline was 43x (s.d. 9.3x) (Supplementary Fig. 8b). The average mtDNA depth of tumour sample was 27,119x (s.d. 13,642x) and the average mtDNA depth of germline was 3,549x (s.d. 2,452x) (Supplementary Fig. 8c).

## Inferencing ancestry from nuclear genome sequencing data

Broad genetic ancestries were estimated using ethnicities from the 1000 genomes project phase 3 (1KGP3)[51] as the truth, by generating PCs for 1KGP3 samples and projecting all participants onto these. We included five broad super-populations: African (AFR), Admixed American (AMR), East Asian (EAS), South Asian (SAS) and European (EUR). The brief steps were as follows: (1) all unrelated samples were selected from the 1KGP3, (2) we selected 188,382 high quality SNPs in our dataset, (3) we further filtered for MAF > 0.05 in 1KGP3 (as well as in our data), (4) we calculated the first 20 principal components using GCTA[54], (5) we projected the individual data onto the 1KGP3 principal component loadings, (6) we trained a random forest model to predict

ancestries based on (i) first 8 1KGP3 principal components, (ii) set Ntrees = 400, (iii) train and predict on 1KGP3 AMR, AFR, EAS, EUR and SAS super-populations. The full details can be found at https://research-help. genomicsengland.co.uk/display/GERE/Ancestry+inference. Genetic ancestry was also predicted and checked using our previously published method[50]. The individuals who were not assigned to any of 5 super-populations were labelled as 'OTHER'. We predicted 1,280 AFR, 170 AMR, 342 EAS, 5,758 SAS, 42,202 EUR and 3,363 OTHER in this study (Fig. 2a). In the cancer germline genomes, we included 312 AFR, 17 AMR, 71 EAS, 338 SAS, 8,348 EUR and 314 OTHER (Extended Data Fig. 6c,d).

We performed a uniform manifold approximation and projection (UMAP)[55] based on the NUMTs which were unique to each population in rare disease genomes. UMAP was analysed using the UMAP package with default parameters in R and visualized using the M3C package[56] in R.

## Extracting mitochondrial DNA sequences and detecting variants

The subset of sequencing reads which aligned to the mitochondrial genome were extracted from each WGS BAM file using Samtools[57]. We ran MToolBox (v1.0)[58] on the resulting smaller BAM files to generate the re-aligned mtDNA BAM files. The re-aligned BAM files were used to call the variants. We also used the second variant caller VarScan2[59] to call mtDNA variants from the re-aligned BAM files (--strand-filter 1, --min-var-freq 0.001, --min-reads2 1, --min-avg-qual 30). The mpileup files used in VarScan2 were generated by Samtools with options -d 0 -q 30 -Q 30. The allele fractions were extracted from VarScan2. We retained only single nucleotide polymorphisms (SNPs) with more than 2 reads on each strand for the minor allele. Variants falling within low-complexity regions (66–71, 300–316, 513–525, 3106–3107, 12418–12425 and 16182–16194) were excluded.

Mitochondrial DNA haplogroup assignment was performed using HaploGrep2[60,61].

## Detecting NUMTs and breakpoints not present in the reference sequence

To detect NUMTs, we used a previously published and validated method[5,15]. From the aligned WGS BAM files we extracted the discordant read pairs using samblaster[62] and included the read pairs where one end aligns to nuclear genome and the other end aligns to the mtDNA reference sequence. The reads with mapping quality equal to zero were discarded. The discordant reads were then clustered together based on sharing the same orientation and whether they were within a distance of 500 bp. We detected the clusters supported by at least two pairs of discordant reads, and filtered out the clusters supported by less than five pairs of discordant reads in our main analysis. The NUMTs within a distance of 1,000 bp on both nuclear DNA and mtDNA were grouped as the same NUMT. We generated two sets of NUMTs based on the NUMTs supported by at least two pairs of discordant reads and at least five pairs of discordant reads (Supplementary Table 1). We observed a weak correlation of the average number of NUMTs and WGS depth ($R^2 = 0.134$, $P < 2.2 \times 10^{-16}$) and mitochondrial genome depth ($R^2 = 0.092$, $P < 2.2 \times 10^{-16}$) (Supplementary Figs. 9a,b) indicating that, although some NUMTs may be missed due to low depth, they are unlikely to have an impact on our conclusions. There was no detected difference of the number of detecting reads with the frequency of NUMTs, suggesting the detection of NUMTs were not biased by the sequencing quality (Supplementary Fig. 9c).

To identify putative breakpoints spanning nuclear DNA and a mtDNA-derived sequence (nuclear-mtDNA breakpoints), we searched for the split reads within a distance of 1,000 bp of discordant reads which were then re-aligned using BLAT[63]. We further analysed the re-aligned reads where one end of the read mapped to nuclear DNA and the other end of the same read mapped to mtDNA-derived sequence. We defined the breakpoints by at least three split reads within the same NUMT. Each NUMT should have one nuclear breakpoint and two mitochondrial breakpoints, with the exception of NUMTs occurring with other nuclear genome structure variations. The breakpoints with 200 bp flanking regions on nuclear genome were annotated using gencode v29[64], gnomAD for pIL scores[65] and a list of datasets were downloaded from UCSC[66] and the publications (see details below). When the NUMTs were involved in multiple genes, we kept the genes with the highest pIL score. The breakpoints on the mitochondrial genome were annotated using MitoMap[67].

## Detecting concatenated NUMTs

To detect putative concatenated NUMTs, first we searched for the breakpoints spanning two locations on the mtDNA-derived sequence (mtDNA–mtDNA breakpoints). We extracted the split reads which only aligned to mtDNA sequence. Those split reads were further re-aligned using BLAT. We analysed the reads where the two ends of the same read mapped to two locations on the mtDNA sequence. We then filtered the breakpoints as follows: (1) each breakpoint had at least 3 split reads observed in at least one individual, (2) each breakpoint had at least 2 split reads observed in the same individual, (3) we excluded the split reads mapped to nearby the start and end of mtDNA genome (the beginning and end of D-loop region), (4) we excluded two concatenated positions less than 50 bp away (they may be mtDNA deletions). Note our method had its limitations—we were not able to separate mtDNA–mtDNA breakpoints within NUMTs from true mtDNA if the breakpoints located around the beginning and end of D-loop region. Thus, our analysis likely missed the concatenated NUMTs where mtDNA–mtDNA breakpoints around the beginning and end of D-loop region. However, our aim was to detect confident concatenated NUMTs and show concatenated NUMTs exist in the humans. After applying the stringent filtering (above), we detected 8,686 breakpoints from 151 different mtDNA–mtDNA breakpoints in 8,450 individuals (Extended Data Fig. 3d). 279 out of 8,686 breakpoints (140 different breakpoints) from 148 individuals were ultra-rare (frequency < 0.1%). One breakpoint (12867–14977) was exceptionally common (frequency 38.4%), which was also commonly seen in an independent dataset in our previous study[5]. To confirm mtDNA–mtDNA breakpoints from the nuclear genome, we performed two independent analyses: (1) we compared the mtDNA–mtDNA breakpoints observed in the offspring and their two parents. If the mtDNA–mtDNA breakpoints were present in the offspring and their fathers, but not in their mothers, we defined them as father-transmitted mtDNA–mtDNA breakpoints. If the mtDNA–mtDNA breakpoints were present in the offspring and their mothers, but not in their fathers, we defined them as mother-transmitted mtDNA–mtDNA breakpoints. Note we were not able to identify the transmission patterns if the mtDNA–mtDNA breakpoints were present in all three family members using the short-read sequencing technique. (2) For the rare and ultra-rare mtDNA–mtDNA breakpoints ($F < 1\%$), we checked whether the individuals carrying the same mtDNA–mtDNA breakpoints also carried the same NUMT.

## Comparing to known NUMTs

Known NUMTs were downloaded from UCSC and previous publications[16–19]. Bedtools[49] was used to search for the known NUMTs in our dataset. Using a conservative approach, we defined the NUMTs as known providing the known NUMTs within 1,000 bp NUMT flanks (upstream 500 bp + downstream 500 bp) detected in this study on the nuclear genome, regardless of the fragments of inserted mtDNA sequences.

## Enrichment analysis

For the enrichment analysis on both nuclear and mtDNA genomes, we studied 1,637 different confident NUMTs with at least 5 discordant reads using a 2-tailed permutation test. Genomics duplications, simple repeats, dbRIP_HS-ME[90], regulatory elements, CpG islands, satellites, retrotransposons (including LINEs and SINEs) and TSS were

downloaded from UCSC[66] (https://genome.ucsc.edu/). Using this information to compute the frequency of each dataset in 200 bp NUMT flanks (upstream 100 bp + downstream 100 bp). Empirical $P$ values were calculated by resampling 1,000 sets of random positions matched to observed NUMTs. For the enrichment on each nuclear genome chromosome, we excluded the Y chromosome due to the complex duplicated structure of Y chromosome sequences limiting confident alignment.

To investigate the relationship between different chromosomes and NUMTs, we applied linear regression in R (http://CRAN.R-project.org/)[68].

$$\text{lm}(\text{Nnumt} \sim \text{Lchr} + \text{Pcentro} + \text{Pcpg} + \text{Pline} + \text{Pltr} + \text{Pretroposon}$$
$$+ \text{Psine} + \text{Pmicrosat} + \text{Prmsk} + \text{Prepeats} + \text{Pdups} + \text{Preg})$$

where Nnumt is number of NUMTs detected in each chromosome, Lchr is the length of chromosome, Pcentro, Pcpg, Pline, Pltr, Pretroposon, Psine, Pmicrosat, Prmsk, Prepeats, Pdups and Preg are $\log_2$-transformed proportions of centromere, CpG islands, LINES, LTRs, retroposon, SINEs, microsatellites, repeats, simple repeats, genomics duplications and regulatory elements on each chromosome.

## Comparing NUMTs with mitochondrial DNA deletions
To study the relationship between NUMT insertion and mitochondrial deletion, we compared the frequency of NUMT breakpoint with the frequency of mitochondrial DNA deletion breakpoint. A list of 1,312 mtDNA deletions were downloaded from mitoBreak database[69]. We calculated the frequencies of breakpoints in different mtDNA regions—D-loop, 13 coding genes, 2 RNAs and combined 22 tRNAs, and compared the distribution with the distribution of breakpoints for germline and tumour-specific NUMTs using linear regression.

## Searching for de novo NUMTs in rare disease trios and tumour-specific NUMTs in cancer genomes
We used the most conservative methods to define the de novo NUMTs from father–mother–offspring trios. We only included NUMTs with at least five pairs of discordant reads in the offspring and none of discordant read detected in the parents.

We applied for the same approach to define tumour-specific NUMTs in cancer genomes. Tumour-specific NUMTs were defined by at least five pairs of discordant reads in the tumour samples and none of discordant reads in the matched normal samples. Lost NUMTs in cancer genomes were defined by at least five pairs of discordant reads in the normal samples and no more than one pair of discordant reads in the matched tumour samples.

## Estimating the rate of de novo NUMTs in trios and tumour-specific NUMTs in cancer genomes
De novo NUMT insertion rate in trios and cancer genomes was estimated as follows:

$$\rho(\text{germline}) = \text{NumtTtrio/Ntrio}$$

$$\rho(\text{tumour}) = \text{NumtTumour/Ngenome}$$

where $\rho$(germline) is the rate of de novo NUMT insertion in trios, $\rho$(tumour) is the rate of tumour-specific NUMT insertion in tumour samples, NumtTtrio is the number of de novo NUMT event in trios, NumtTumour is the number of tumour-specific NUMTs, Ntrio is the number of total trios and Ngenome is the number of total normal–tumour pairs.

## Analysing the correlation of tumour-specific NUMTs and cancer types
To understand the relationship between donor age, sex and the average number of NUMTs, we applied linear regression to each dataset using R (http://CRAN.R-project.org/).

$$\text{Model 1} < -\text{lm}(N \sim \text{Age} + \text{Sex} + \text{DPmt})$$
$$\text{Model 2} < -\text{lm}(\text{Nsoma} \sim \text{Age} + \text{Sex} + \text{DPmt})$$

Where $N$ and Nsoma are average numbers of NUMTs and tumour-specific NUMTs, Age is donor age, Sex is donor sex and DPmt is average mitochondrial DNA sequencing depth.

## Detecting cancer SNVs, indels and structural variants
Read alignment against human reference genome GRCh38-Decoy+EBV was performed with ISAAC (version iSAAC-03.16.02.19)[70], SNVs and short insertions–deletions (indels) variant calling together with tumour − normal subtraction was performed using Strelka (version 2.4.7)[71]. Strelka filters out the following germline variant calls: (1) all calls with a sample depth three times higher than the chromosomal mean, (2) site genotype conflicts with proximal indel call, (3) locus read evidence displays unbalanced phasing patterns, (4) genotype call from variant caller not consistent with chromosome ploidy, (5) the fraction of basecalls filtered out at a site > 0.4, (6) locus quality score < 14 for heterozygous or homozygous SNP, (7) locus quality score < 6 for heterozygous, homozygous or het-alt indels, (8) locus quality score < 30 for other small variant types or quality score is not calculated. Strelka filters out the following somatic variant calls: (1) all calls with a normal sample depth three times higher than the chromosomal mean, (2) all calls where the site in the normal sample is not a homozygous reference, (3) somatic SNV calls with empirically fitted VQSR score < 2.75 (recalibrated quality score expressing the phred scaled probability of the somatic call being a false positive observation), (4) somatic indels where fraction of basecalls filtered out in a window extending 50 bases to either side of the indel's call position is > 0.3, (5) somatic indels with quality score < 30 (joint probability of the somatic variant and a homo ref normal genotype), (6) all calls that overlap LINE repeat region.

Structural variants (SVs) and long indel (>50 bp) calling was performed with Manta (version 0.28.0)[72] which combines paired and split-read evidence for SV discovery and scoring. Copy number variants (CNVs) were called with Canvas (version 1.3.1)[73] that employs coverage and minor allele frequencies to assign copy number. These tools filter out the following variant calls: (1) Manta-called SVs with a normal sample depth near one or both variant break-ends three times higher than the chromosomal mean, (2) Manta-called SVs with somatic quality score < 30, (3) Manta-called somatic deletions and duplications with length > 10kb, (4) Manta-called somatic small variant (<1kb) where fraction of reads with MAPQ0 around either break-end > 0.4, (5) Canvas-called somatic CNVs with length < 10kb, (6) Canvas-called somatic CNVs with quality score < 10. The full details of bioinformatics pipeline can be found at https://research-help.genomicsengland.co.uk/pages/viewpage.action?pageId=38046624.

## Searching for the evidence of the mechanism of NUMT insertions
**PRDM9.** PRDM9 determines the locations of meiotic recombination hotspots where meiotic DNA DSBs are formed. To investigate the mechanism of NUMT insertions, we compared the NUMTs with a set of 170,198 published PRDM9-binding peaks cross the genome[74]. We counted the number of NUMTs overlapping PRDM9-binding peaks and performed the permutation analysis (see the details in 'Enrichment analysis'). Next, we calculated the distance between the breakpoint of each NUMT (from both the germline and tumour-specific NUMTs) with the nearest PRDM9-binding site.

**Human DNA repair genes.** A list of known human DNA repair genes was downloaded from Human DNA Repair Genes website (https://www.mdanderson.org/documents/Labs/Wood-Laboratory/human-dna-repair-genes.html)[38,39]. We extracted the somatic missense mutations in DNA repair genes from all cancer samples, and compared the relationship between samples carrying the mutations and tumour-specific NUMTs.

**Somatic mutational signatures.** Somatic mutation signatures are the consequence of multiple mutational processes that the human body is subjected to throughout life. Each different process generates a unique combination of mutation types that are called mutation signatures (https://cancer.sanger.ac.uk/signatures/signatures_v2/). Mutational signature was computed using the R package nnls (https://CRAN.R-project.org/package=nnls). The details of how the signatures were computed is described in Alexandrov et al., 2013[75] and online document https://research-help.genomicsengland.co.uk/pages/viewpage.action?pageId=38046624.

## Assessing clinical significance

**Rare disease participants with no known genetic diagnosis.** The Genomics England PanelApp (https://panelapp.genomicsengland.co.uk/)[76] list of genes and genomic entities were used to provide a list of potential disease genes (*N* = 5,883). NUMTs were identified that had a frequency of < 1%, and their breakpoints within 200 bp flanking regions of one of these genes. Consequence annotation was done with gencode v29, including gene, intron, exon, CDS, start codon, stop codon, five prime UTR and three prime UTR regions[64]. NUMTs which were annotated as falling in an exon were analysed in detail. For each gene, we considered the strength of evidence that the gene is associated with a disease, the inheritance pattern of the disorder, the reported types of pathogenic variants and reported mechanism of disease (for example, haploinsufficiency, gain of function or repeat expansion), using information from OMIM (https://omim.org/)[77] and by searching PubMed (https://pubmed.ncbi.nlm.nih.gov/). For the established disease genes, we considered available clinical information for each proband which included their Human Phenotype Ontology terms[91], family history and age at enrolment. We assumed that the rare NUMT was present on one allele only, unless it was present in both parents or there was documented consanguinity (where parental data was not available). For recessive disorder genes containing a NUMT, we looked whether it was present in one or both parents (if available), whether there was a family history of consanguinity, and at the sequence data to see whether there was a second rare variant. The location of the NUMT insertion was explored in UCSC genome browser[66].

**Rare disease participants with a genetic diagnosis.** Participants with a confirmed genetic diagnosis were identified from the Genomic Medicine Centre exit questionnaire (https://research-help.genomicsengland.co.uk/pages/viewpage.action?pageId=38046767). Genomic coordinates of the causative variant were compared with the genomic coordinates of the NUMTs using bedtools[49].

**Rare disease NUMTs in participants with mitochondrial DNA maintenance disorders.** Participants with mitochondrial DNA maintenance disorders[78] were identified from the Genomic Medicine Centre exit questionnaire and from our previous analysis of participants with suspected mitochondrial disorders[79]. We also identified affected family members who had genome sequencing data available. 122 NUMTs were detected from 20 individuals. Only 4 NUMTs (2 different NUMTs) from two families in exons. We compared the average number of NUMTs in these participants to the rest of the rare disease participants.

**Cancer genomes.** To determine whether a NUMT insertion was a driver mutation in the development of cancers, NUMTs with 200 base pairs flanking region were identified which were located genes of interest. Our genes of interest were defined as those on the COSMIC (Catalogue of Somatic Mutations in Cancer) Cancer Gene Census list (tier 1 and tier 2) which includes genes known to contain mutations causally implicated in cancer[28]. We also used a list of known human DNA repair genes[38,39]. The location of the NUMT insertion in relation to these gene lists was explored in the UCSC genome browser.

## Validating the NUMTs using long-read sequencing

To validate NUMT detection in short-read sequencing, we carried out whole-genome sequencing on Oxford Nanopore PromethION in 39 individuals from rare disease genomes. To maximize sequencing yield, 4 µg of germline DNA from 100KGP participants was fragmented to 15–30 Kb with Covaris g-tubes (4,000 rpm, 1 min, 1–3 passes until the desired length was achieved) and then depleted of low molecular weight DNA (<10 Kb) with the Short Read Eliminator kit (Circulomics, SS-100-101-01) as described by the manufacturer. After checking DNA size distribution on an Agilent Femto Pulse system, a sequencing library was generated with the Oxford Nanopore SQK-LSK109 kit, starting from 1.2 µg of high molecular weight-enriched DNA. Samples were quantified with a Qubit fluorometer (Invitrogen, Q33226) and 500 ng loaded onto a PromethION R.9.4.1 flow cell following manufacturer's instructions. In experiments where throughput was limited by a rapid increase in unavailable pores, the library was re-loaded following a nuclease flush ~20hrs after the initial run. Base-calling was performed with Guppy-3.2.6/3.2.8 in high accuracy mode. Full details of the protocol can be found at https://research-help.genomicsengland.co.uk/display/GERE/Genomic+Data+from+ONT?preview=/38046759/38047942/v1_protocol_ONT_LSK109.pdf. Sequencing reads were aligned to GRCh38 using minimap2[80] version 2.17. QC statistics and plots were generated using Nanoplot[81] version 1.26.0. The full details of bioinformatics pipeline can be found at https://research-help.genomicsengland.co.uk/display/GERE/Genomic+Data+from+ONT?preview=/38046759/38047944/PromethION%20SV%20calling%20pipeline%20GRCh38.docx. We then extracted the long reads aligned to the same region where a NUMT detected using short-read sequencing from the same individual. The extracted long reads were re-aligned using BLAT. The observed NUMTs were also manually inspected on Integrated Genomics Viewer (IGV)[82]. 182 out of 184 NUMTs (29 out of 31 distinct NUMTs) detected using short-read sequencing were also seen in long-read sequencing data. Two NUMTs from the same individual were missing in long-read sequencing likely due to the low number of aligned reads in long-read sequencing.

## Detecting methylation state of NUMTs using long-read sequencing

Whole-genome-wide methylation detection was carried out using call-methylation function from Nanopolish v0.13.3[83] in 39 individuals. The methylation detection output includes the position of the CG dinucleotide on the reference genome, the ID of the read that was used to make the call, and the log-likelihood ratio. We extracted the long reads mapped to mtDNA genome, and further grouped them into two groups: (1) long reads also mapped to nuclear genome, (2) long reads only mapped to mtDNA genome. Next, we calculated methylation frequency of each site using the calculate_methylation_frequency.py script from the package in each read group. The methylation calls detected by the 1st group were from NUMTs, and the calls detected by the 2nd group were from true mtDNA. We used the methylation profile of true mtDNA as reference, and NUMTs methylation was estimated as the $\log_2$ ratio of methylation frequency of each site between NUMTs and true mtDNA from the same individual. Note, if the individuals carried concatenated NUMTs, the calls detected by 2nd group were from mixed true mtDNA and concatenated NUMTs. We were not able to separate the long reads mapped to the middle of concatenated NUMTs where the reads also only mapped to mtDNA genome and true mtDNA genome.

In this analysis, we focused on the concatenated NUMTs and the large NUMTs where long reads were confidently aligned to NUMTs. We only included the calls with at least 3 reads mapped to NUMTs and at least 10 reads mapped to true mtDNA sequences. We also used 4 reads, 5 reads, 6 reads, 7 reads, 8 reads 9 reads and 10 reads as the cut-offs to detect NUMTs methylation. We observed the same distribution of methylation frequency across different cut-offs (Fig. 3a), indicating read-thresholds did not affect our results.

## Detecting mutations within the NUMT insertions

We performed a de novo assembly of all 335,891 NUMTs detected in this study. The steps of processes were: (1) we clustered the discordant reads detected from each NUMT in the same individual. (2) The consensus sequence of NUMT contig was generated using CAP3[84]. (3) The contigs were then aligned against mitochondrial reference genome[85] using Blat[63] and Clustal Omega[86]. (4) The aligned sequences from Clustal Omega were used to detect the nucleotide changes between NUMT sequences and mitochondrial reference genome sequences using BioPython[87]. To ensure the confident calls, we applied the additional filtering as follows: (1) we only included NUMTs shorter than 1,000 bp; (2) we excluded the variants within 5 bp of NUMT breakpoints; (3) we removed the variants where the aligned reference allele were different from mtDNA reference genome at the same position; (4) we only included single nuclear variations; (5) we excluded the individuals carrying many more variants than the overall population (> mean number of variants + 3 × s.d.).

To define NUMT-specific variants, we applied the additional filtering: (1) we excluded variants present more than 50% individuals carrying the same common or rare NUMTs and 75% individuals carrying the same ultra-rare NUMTs. This stringent filtering strategy was designed to provide maximum confidence that any NUMT-specific variants were highly likely to have occurred after NUMT sequences have inserted into nuclear genome, compromising the sensitivity of the analysis. (2) We excluded variants only detected in 1 individual to minimize the likelihood of sequencing errors; (3) to obtain the most confident NUMT-specific mutations, we only included the variants detected in at least two individuals from the same family. In the main text, we reported 3 groups of NUMT-specific variants. Total group A, after applying step (1); subgroup B, after step (2); and subgroup C, after step (3).

## Estimating the ages of NUMTs

The age of NUMTs was estimated using the method described previously[19]. We aligned the mitochondrial sequences from human, chimpanzee and the consensus sequence from each NUMT contig using Clustal Omega. The ancestral mitochondrial sequences from chimpanzee was downloaded from ENSEMBL(Pan_tro_3.0). The aligned sequences were used to generate the nucleotide changes using BioPython. We calculated the ratio of the number of sites that matched human allele to the total number of sites where the human and ancestral mitochondrial sequences differ within each NUMT region. The ratio was used to derive an approximate age for each NUMT, relative to an estimated human-chimpanzee divergence time of 6 million years. To ensure the confident results, we applied the filtering as follows: (1) we only included NUMTs with length between 50 and 1,000 bp; (2) we excluded NUMTs without different allele between human and chimpanzee; (3) the age was estimated from more than 50% of individuals carrying the same NUMT and at least in 2 individuals. After applying this filtering, we excluded all the private NUMTs which were only seen in one individual. (4) We excluded concatenated NUMTs.

## Statistical analysis and plotting

All statistical analyses in this study were suggested in the text and performed using R[68] (http://CRAN.R-project.org/) and Python (http://www.python.org). Figures were generated using R and Matplotlib (https://matplotlib.org) in Python. Circos plots were made using Circos (http://circos.ca/)[88]. Chromosome maps were made using chromoMap[89].

A web interface to deposit NUMTs detected in this study was developed using Shiny v1.7.1 (https://CRAN.R-project.org/package=shiny) (https://cran.r-project.org/web/packages/shiny/index.html)[92].

## Web resources

NUMTs detected in this study are publicly available through a web interface at https://wwei.shinyapps.io/numts/.

## Reporting summary

Further information on research design is available in the Nature Research Reporting Summary linked to this article.

## Data availability

WGS data from the participants enrolled in 100,000 Genomes Project can be accessed via Genomics England Limited following the procedure outlined at: https://www.genomicsengland.co.uk/about-gecip/joining-research-community/. In brief, applicants from registered institutions can apply to join one of the Genomics England Clinical Interpretation Partnerships, and then register a project enabling, access to the Genomics England Research Environment 2 h after completing online training. *H. sapiens* NCBI GRCh38 assembly can be found at https://www.ncbi.nlm.nih.gov/assembly/. Gencode v29 can be found at https://www.gencodegenes.org/human/release_29.html. Human genome annotation files can be found at https://hgdownload.soe.ucsc.edu/goldenPath/hg38/database/. The ancestral mitochondrial sequences from Chimpanzee can be found at https://www.ensembl.org/Pan_troglodytes/Info/Index.

## Code availability

Code used in the study is available at https://github.com/Wei-Wei060512/NUMTs-detection.git and https://doi.org/10.5281/zenodo.6966017.

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

**Acknowledgements** This study makes use of data generated by the Genomics England 100,000 Genomes Project. The main source of funding for the BioResource and Genomics England is provided by the National Institute for Health Research of England (NIHR, http://www.nihr.ac.uk). This work was also made possible by funding from the UK Medical Research Council (MRC) to create the UK Clinical Genomics Datacentre. P.F.C. is a Wellcome Trust Principal Research Fellow (212219/Z/18/Z), and a UK NIHR Senior Investigator, who receives support from the Medical Research Council Mitochondrial Biology Unit (MC_UU_00015/9), the Medical Research Council (MRC) International Centre for Genomic Medicine in Neuromuscular Disease (MR/S005021/1), the Leverhulme Trust (RPG-2018-408), an MRC research grant (MR/S035699/1), an Alzheimer's Society Project Grant (AS-PG-18b-022) and the National Institute for Health Research (NIHR) Biomedical Research Centre based at Cambridge University Hospitals NHS Foundation Trust and the University of Cambridge. K.R.S. was supported by an MRC strategic award to establish an International Centre for Genomic Medicine in Neuromuscular Diseases (ICGNMD) MR/S005021/1. M.J.C. is an NIHR Senior Investigator and this work forms part of the portfolio of research of the NIHR Barts Biomedical Research Centre. Genomics England and the 100,000 Genomes Project was funded by the National Institute for Health Research, the Wellcome Trust, the Medical Research Council, Cancer Research UK, the Department of Health and Social Care and NHS England. We thank all the participants and healthcare teams who made this study possible. The views expressed are those of the author(s) and not necessarily those of the NHS, the NIHR or the Department of Health and Social Care.

**Author contributions** P.F.C. and W.W. conceived of the overall study. W.W. performed the bioinformatic and statistical analysis with additional contributions from K.R.S. and M. Tischkowitz for the cancer gene analysis, K.R.S. for the rare disease gene analysis, and G.E., A.O., M. Tanguy and A.G. for the long-read sequencing. M.J.C. had a key role in advising and overseeing access to the Genomics England dataset. P.F.C. and W.W. wrote the first draft of the manuscript, which all authors critically edited. P.F.C. supervised the study and sought the funding.

**Competing interests** The authors declare no competing interests.

**Additional information**
**Correspondence and requests for materials** should be addressed to Patrick F. Chinnery.

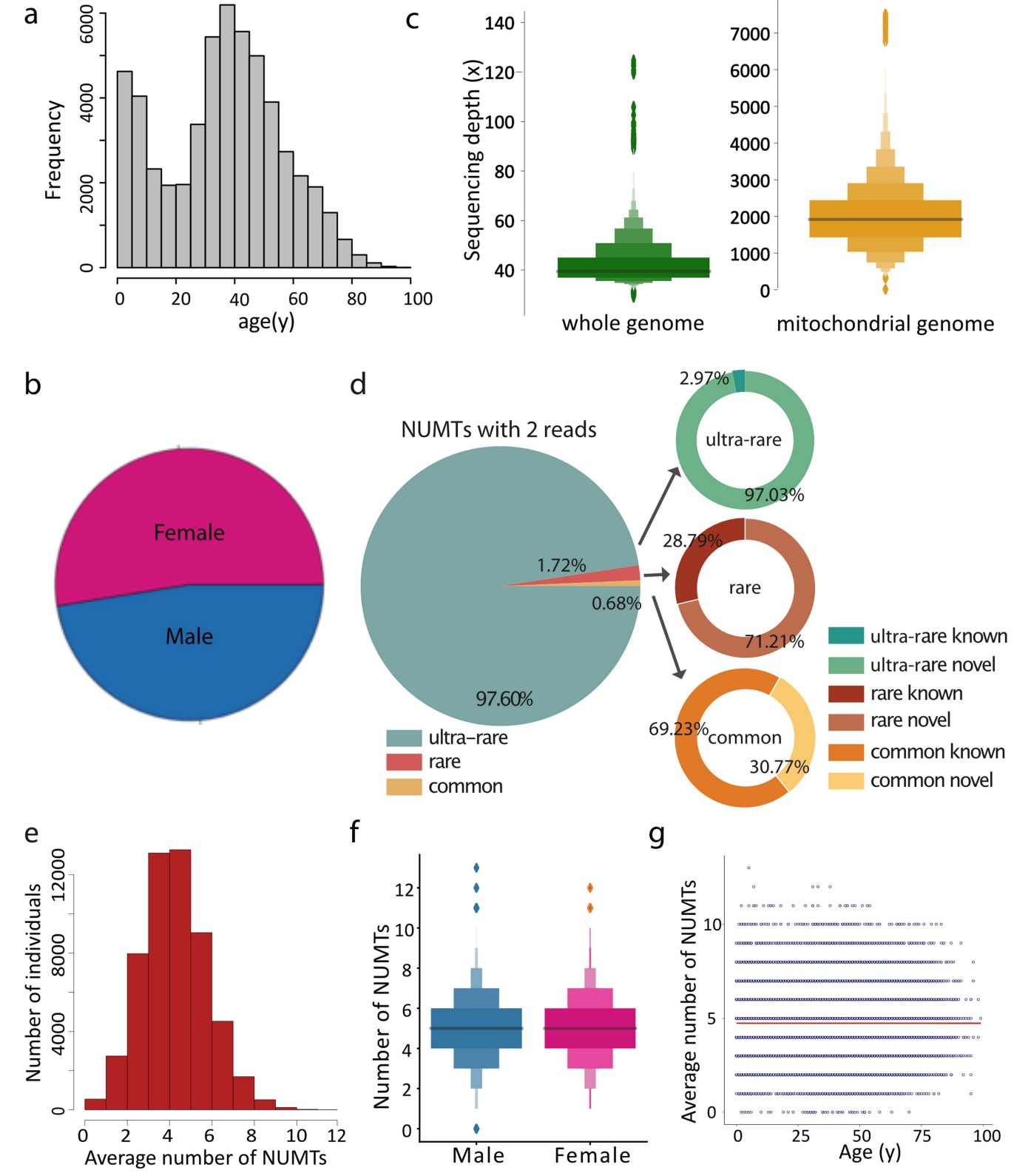

**Extended Data Fig. 1** | See next page for caption.

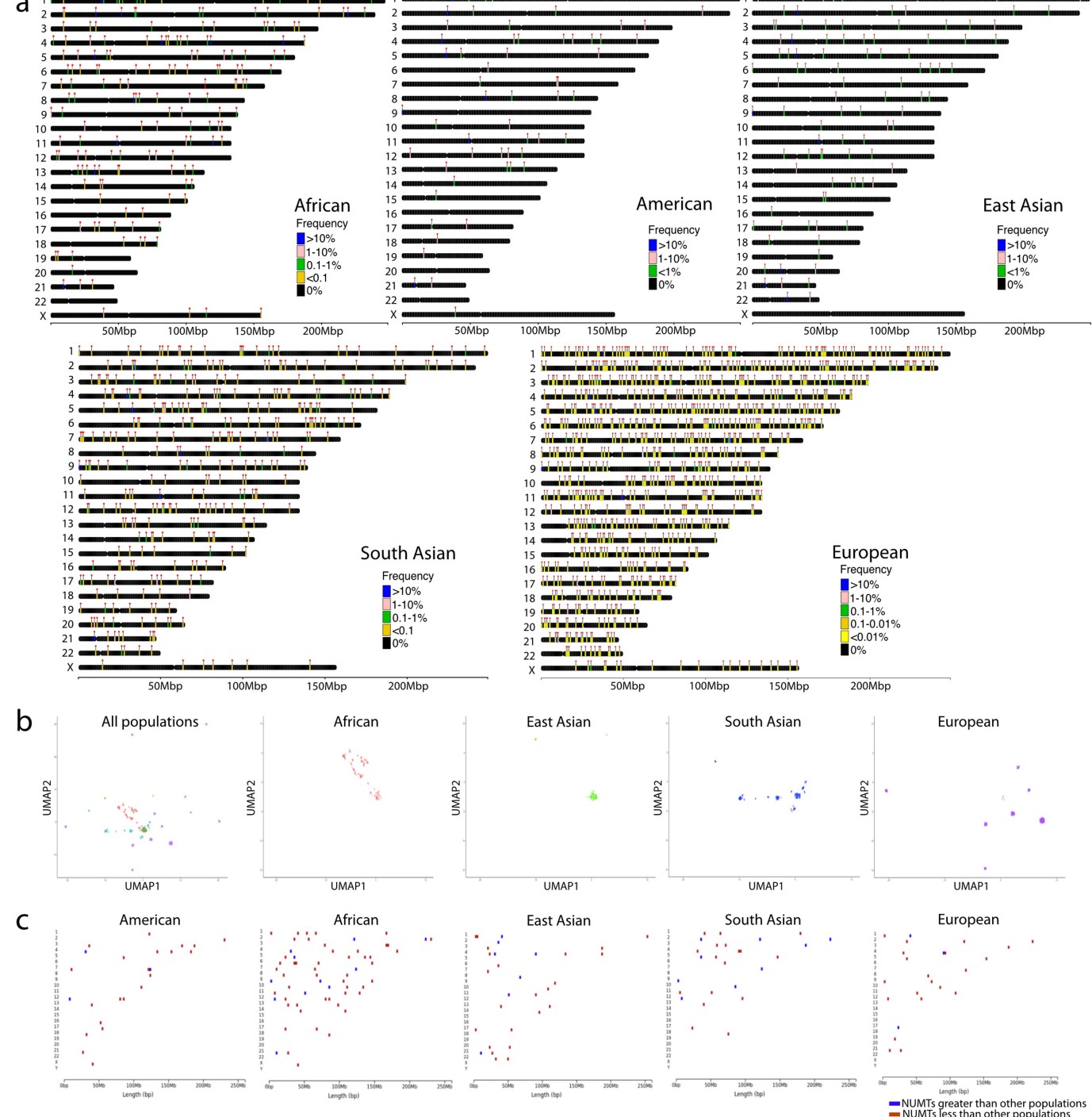

**Extended Data Fig. 2 | NUMTs detected in the different populations.**
**a**. Chromosome map of NUMTs detected in African, American, East Asian, South Asian and European genomes. Chromosomal locations of different NUMT insertions coloured by the frequency (F) of NUMTs. Dots show the locations of the NUMTs. **b**. A uniform manifold approximation and projection (UMAP) of germline NUMTs in all populations and 4 sub-populations. **c**. Chromosomal locations of NUMTs were significantly greater / less detected in the different populations.

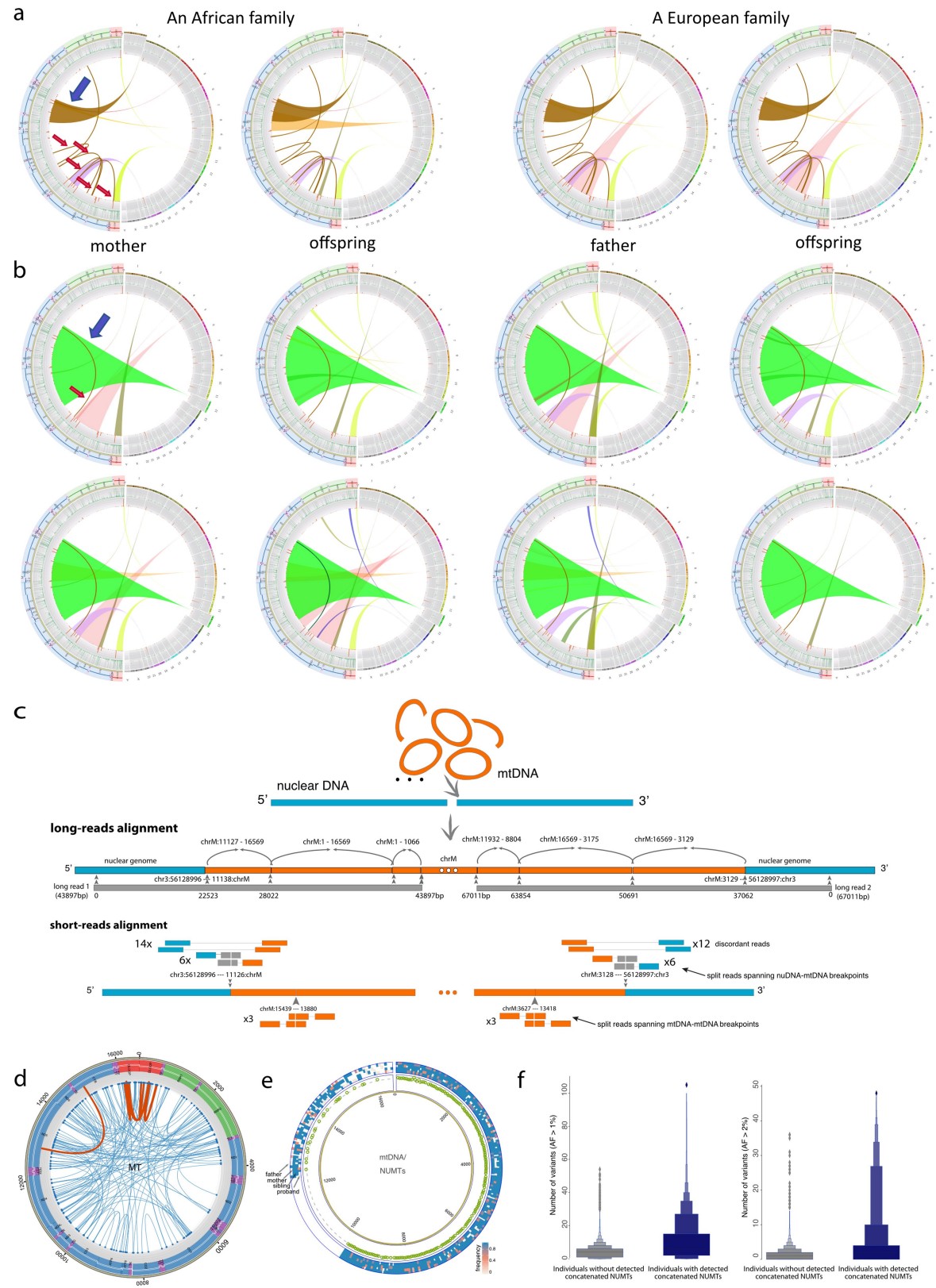

**Extended Data Fig. 3** | See next page for caption.

**Extended Data Fig. 3 | Concatenated NUMTs and long-read sequencing validation. a**. Circos plots show 4 individuals from 2 families shared 5 mtDNA-mtDNA breakpoints which were exclusively present in 4 individuals, and also shared an ultra-rare NUMT insertion which was only seen in the same 4 individuals. **b**. Circos plots show 8 individuals shared 1 mtDNA-mtDNA breakpoint which was exclusively present in these 8 individuals, and also shared a NUMT insertion which was only seen in the same 8 individuals. Blue arrows point to the shared NUMTs. Red arrows point to the shared mtDNA-mtDNA breakpoints. **c**. Model showing the formation of concatenated NUMTs and our strategy for their detection using both long-read sequencing and short-read sequencing. mtDNA and nuclear genome sequences are shown in orange and blue. Reads mapped to both mtDNA and nuclear genome sequences are shown in grey, mapped to only mtDNA sequences are in orange and mapped to only nuclear genome sequences in blue. **d**. Circos plot of mtDNA-mtDNA breakpoints detected in the rare disease genomes. mtDNA-mtDNA breakpoints were detected by split reads mapping only to mtDNA. Complex concatenated NUMTs contain multiple mtDNA fragments. Detection of mtDNA-mtDNA breakpoints support the putative concatenated NUMTs. Common and rare mtDNA-mtDNA breakpoints (frequency > = 0.1%) shown in red links. Ultra-rare mtDNA-mtDNA breakpoints (frequency < 0.1%) shown in blue links. **e**. Circos plot shows the methylation frequency of a rare NUMT (insertion mt.12314 – 9526 bp, frequency = 0.26%) detected in 4 members from the same family (father, mother, sibling and proband). Circles from the outside to the inside indicate the following: (1) methylation frequency of NUMTs detected by split long-reads in father, mother, sibling and proband, (2) ratio of methylation frequency between NUMTs and "true" mtDNA sequences in all 4 family members. Green dots were the sites methylated in NUMTs. Colour key corresponds to the methylation frequencies. **f**. Letter-value plots of the average number of observed mtDNA variants (left – variant frequency > 1%, right - variant frequency > 2%), individuals carrying putative concatenated NUMTs and without putative concatenated NUMTs shown, separately. Variants observed in the individuals carrying putative concatenated NUMTs are mixed variants from both mtDNA sequence and NUMTs. The middle line represents the median (50th percentile). Each successive level outward contains half of the remaining data. The first two sections out from the centre line contain 50% of the data. The next two sections contain 25% of the data. This continues until at the outlier level. The outliers are plotted as diamonds.

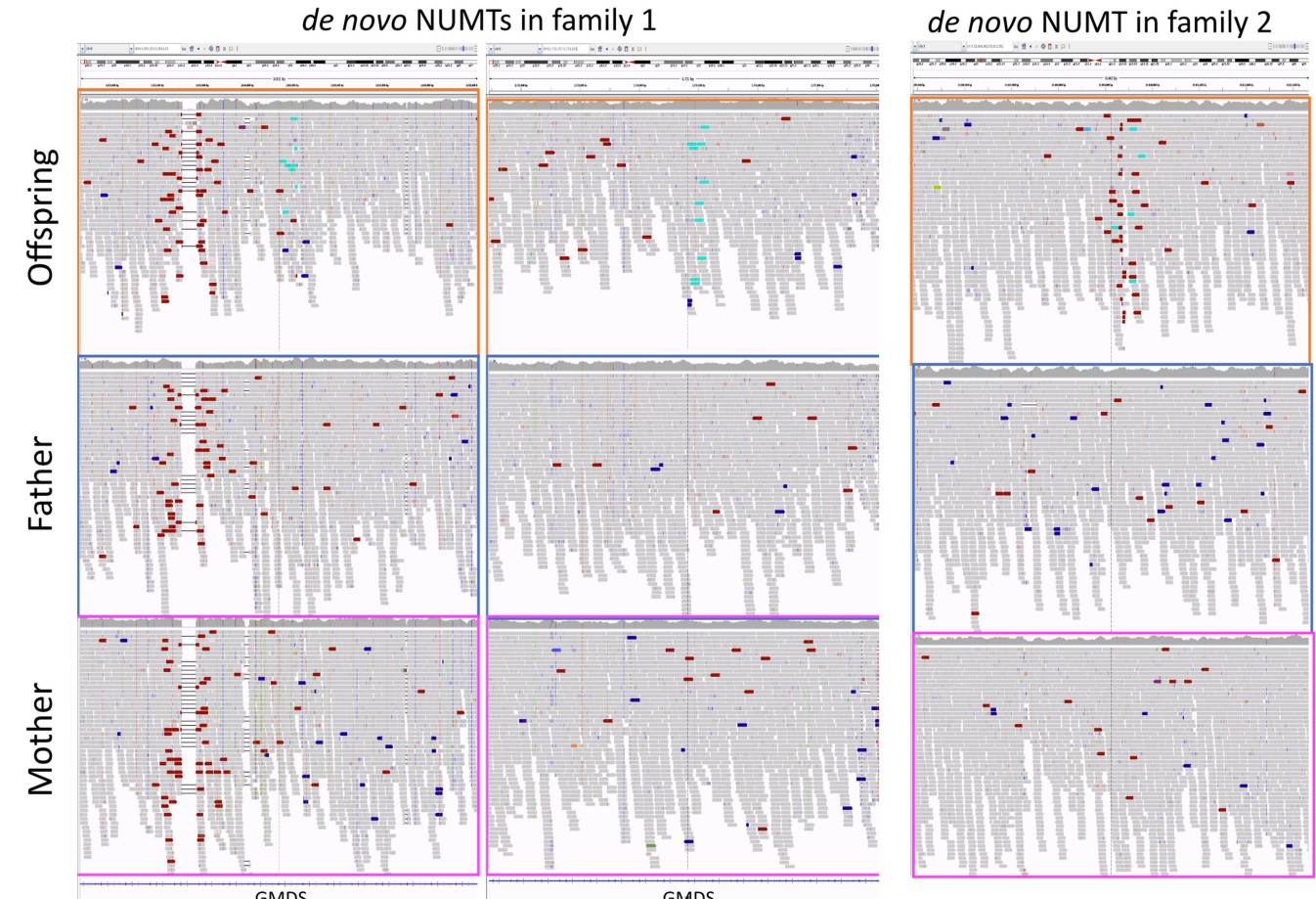

**Extended Data Fig. 4 | IGV alignment of *de novo* NUMTs in the rare disease genomes.** Integrative Genomics Viewer (IGV) screenshots show the aligned reads corresponding to three *de novo* NUMTs observed in two families. Teal bars indicate the aligned reads which mapped to the nuclear DNA where their mates mapped to the mtDNA. In family 1, offspring carried two NUMTs within the same gene, but not seen in either of the parents. In family 2, offspring carried a NUMT which was not seen in either of the parents.

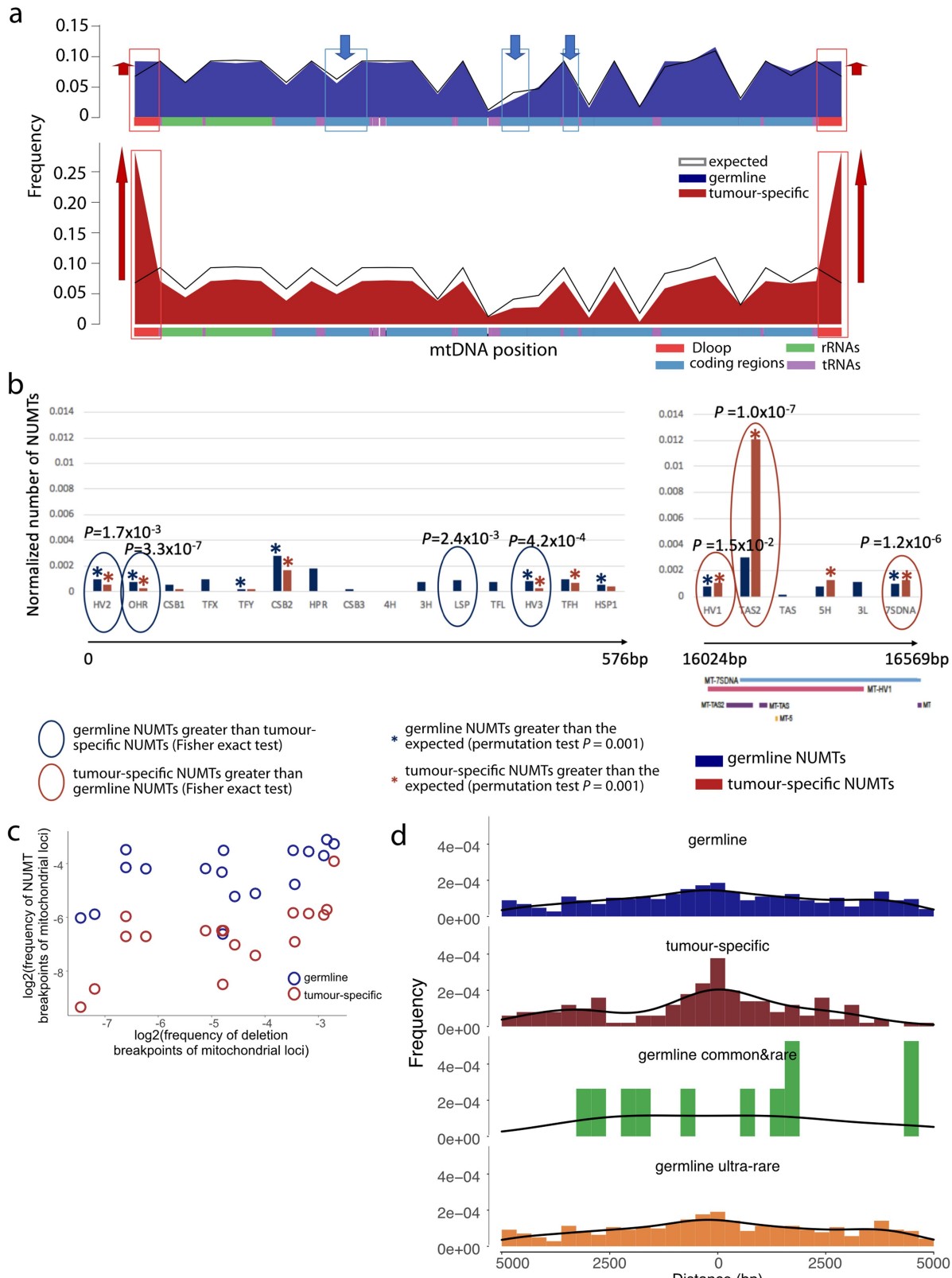

**Extended Data Fig. 5** | See next page for caption.

**Extended Data Fig. 5 | Frequency of NUMT breakpoints on mtDNA genome and the distance of NUMT location to nuclear transcription start sites (TSS). a**. Normalized frequency of NUMT breakpoints in each mtDNA region. Black lines are expected frequency. Top blue area plot shows the frequency of breakpoints from germline NUMTs. Bottom red area plot shows the frequency of breakpoints from tumour-specific NUMTs. Mitochondrial regions are shown in the different colours at the bottom of each plot. Red boxes highlight the regions where the frequencies were significantly greater than the expected by chance. Blue boxes highlighted the regions where the frequencies significantly less than the expected by chance. **b**. Normalized number of NUMTs within each Dloop region. Stars represent the NUMTs were significantly enriched in each region (permutation test). Circles labelled P values were from the comparison of germline and tumour-specific NUMTs (two-sided Fisher's exact test). **c**. Correlation of frequencies of deletion breakpoints and NUMT breakpoints in each mtDNA region from germline and tumour-specific NUMTs. **d**. Histogram of distance of NUMTs location to transcription start sites (TSS). Germline, germline common & rare, ultra-rare and tumour-specific NUMTs are shown, separately.

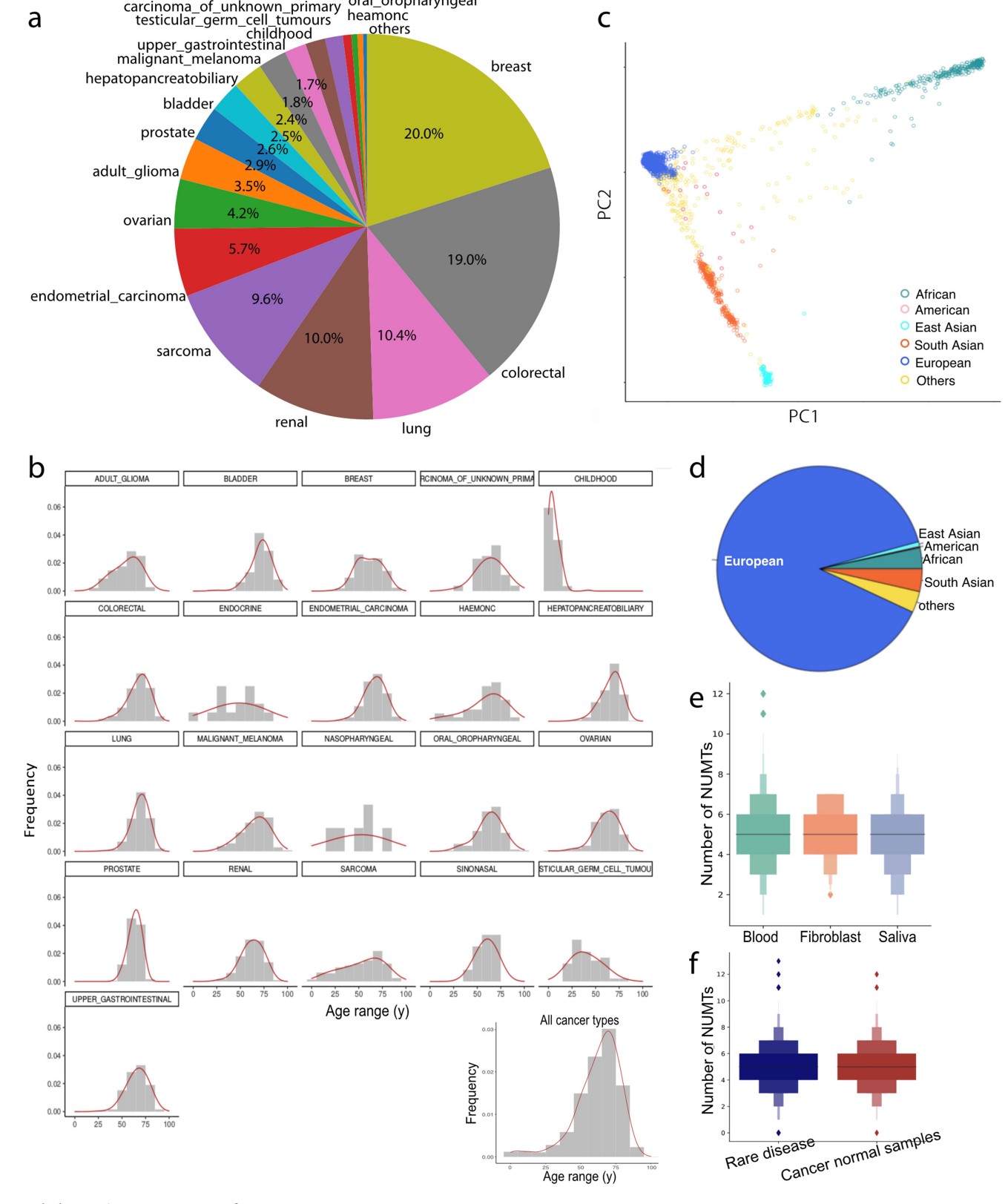

**Extended Data Fig. 6** | See next page for caption.

**Extended Data Fig. 6 | Whole genome sequencing in 12,509 normal-tumour pairs from the Genomics England Cancer Project and detected NUMT insertions. a**. Pie chart of proportion of sample size from each cancer type included in this study. **b**. Histogram of tumour donor age from all cancer types (bottom right) and each cancer type. **c**. Projection of the nuclear genotypes at common SNPs onto the two leading principal components (PC1 and PC2) computed with the 1000 Genomes dataset from the cancer genomes, with individuals coloured by their assigned nuclear ancestry. **d**. Proportion of sample size from each population in the cancer genomes. **e**. Number of NUMTs detected in the different tissue types from the matched normal tissue samples taken from cancer participants. The middle line represents the median (50th percentile). Each successive level outward contains half of the remaining data. The first two sections out from the centre line contain 50% of the data. The next two sections contain 25% of the data. This continues until at the outlier level. The outliers are plotted as diamonds. **f**. Number of NUMTs detected in the rare disease blood samples and the matched normal tissue samples taken from cancer participants. The middle line represents the median (50th percentile). Each successive level outward contains half of the remaining data. The first two sections out from the centre line contain 50% of the data. The next two sections contain 25% of the data. This continues until at the outlier level. The outliers are plotted as diamonds.

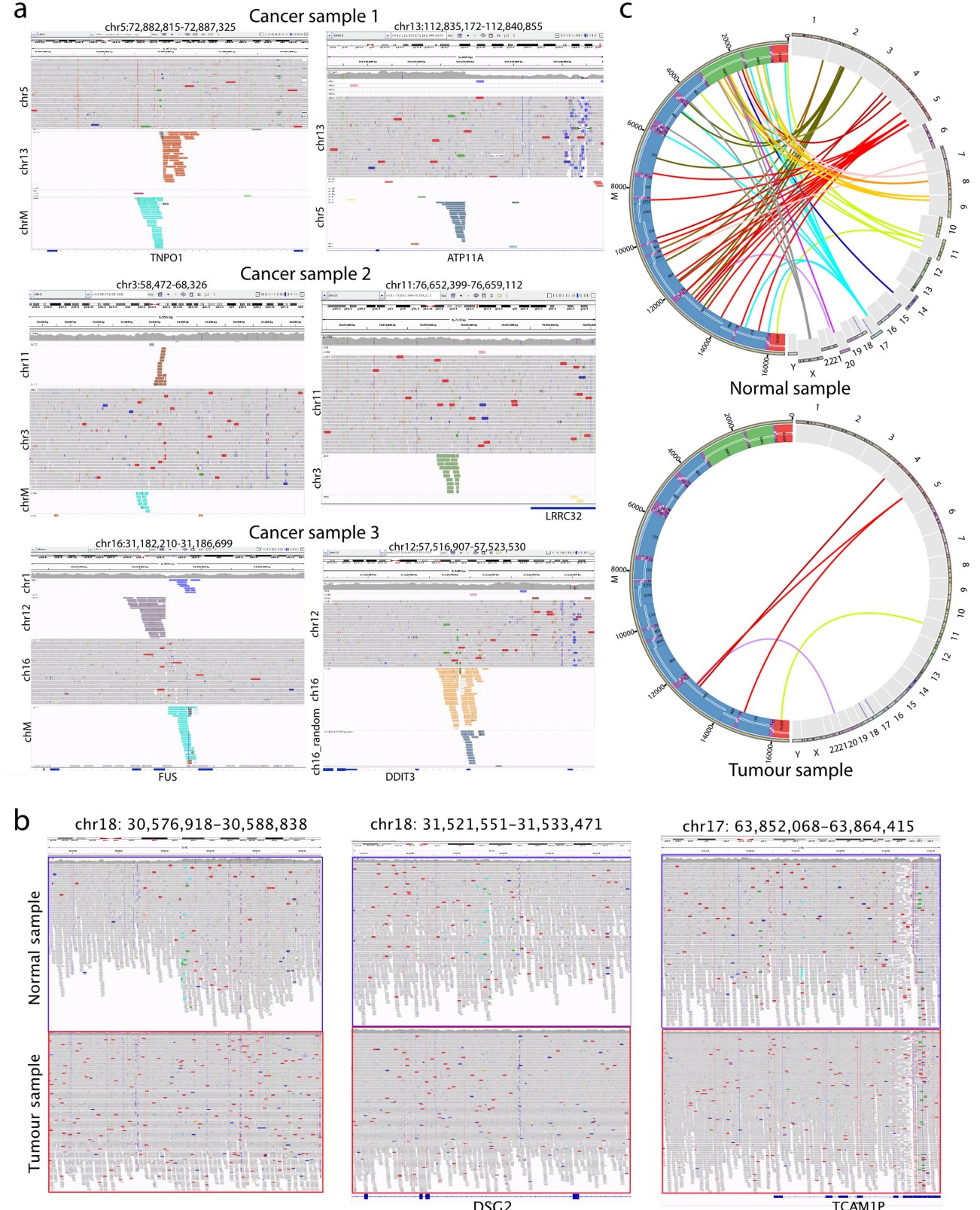

**Extended Data Fig. 7** | See next page for caption.

**Extended Data Fig. 7 | Examples of IGV alignment of NUMTs. a**. Examples of IGV alignment of tumour-specific NUMTs coupled with other translocation variations in the nuclear genome. Teal bars indicate the aligned reads which mapped to the nuclear DNA where their mates mapped to the mtDNA. Other non-grey colour bars indicate the aligned reads which mapped to one nuclear chromosome where their mates mapped to a different nuclear chromosome. For example, Cancer sample 1 had one NUMT (teal bars) on chromosome 5 and another translocation variation between chromosome 5 and chromosome 13 (orange bars) in the same region (left). The same translocation variation was also seen on chromosome 13 (right). The aligned reads mapped to chromosome 13 where their mates mapped to chromosome 5 (steel blue bars). **b**. An example of IGV alignment of tumour lost NUMTs. IGV screenshots show the aligned reads corresponding to the lost NUMTs in one breast tumour sample. Teal bars indicate the aligned reads which mapped to the nuclear DNA where their mates mapped to the mtDNA. NUMTs only present in the matched normal sample but not in the tumour sample, with the average sequencing depth of tumour sample (128x) was more than three-times deeper than the matched normal sample (40x). **c**. Cirocs plot illustrates an example of lost NUMT in a haematological tumour sample. The links represent all NUMTs detected in either normal sample or tumour sample. The tumour sample lost many NUMTs across the whole genome, with the average sequencing depth of tumour sample (116x) was more than twice deeper than the matched normal sample (40x).

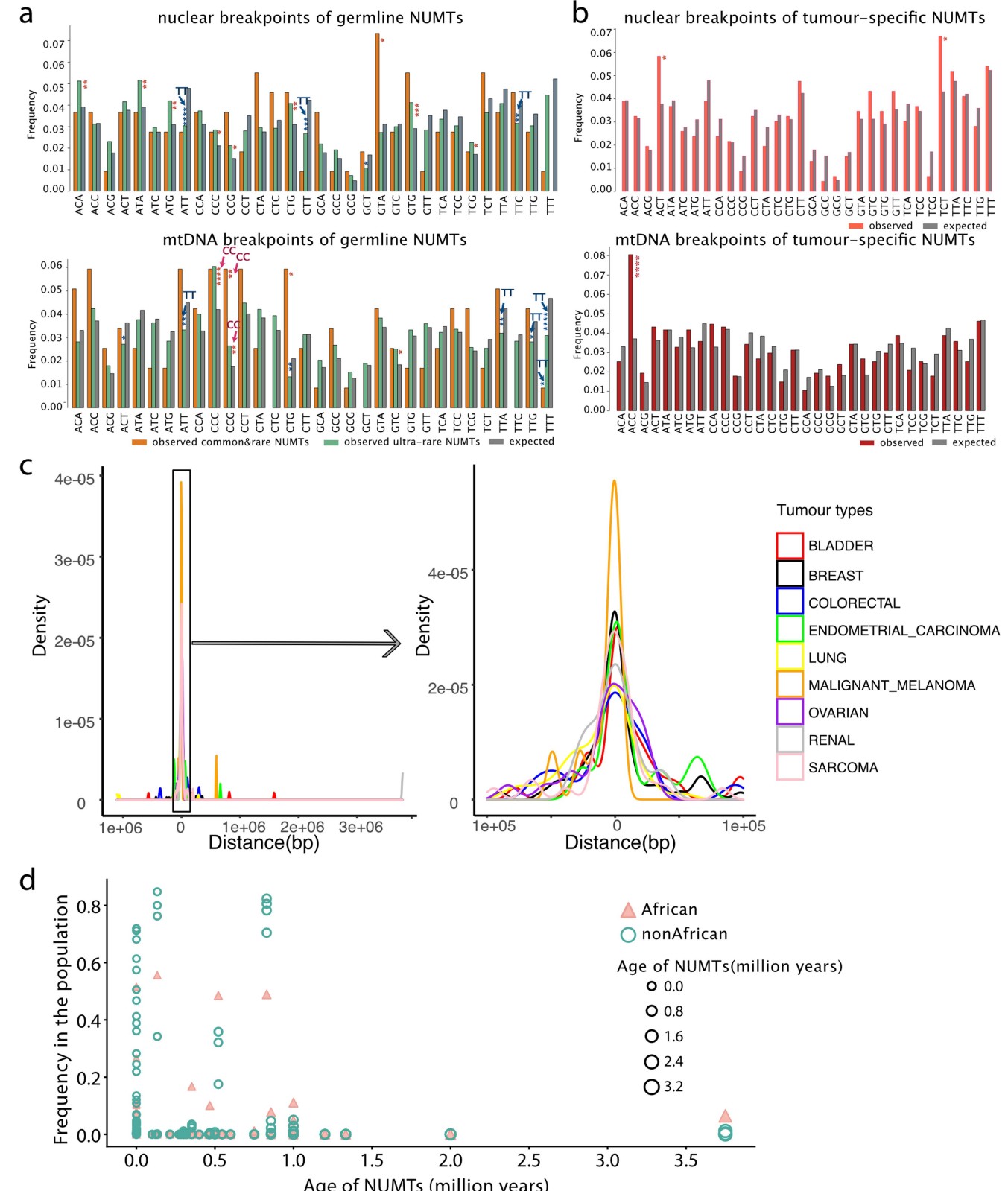

**Extended Data Fig. 8 |** See next page for caption.

**Extended Data Fig. 8 | NUMT nuclear breakpoints, relation to PRDM9 binding sites, and NUMT age. a**. Frequencies of trinucleotides around germline NUMTs breakpoints. The breakpoints of nuclear genome are shown at the top and mtDNA genomes at the bottom, common&rare, ultra-rare NUMTs and the expected frequencies shown in the different colours. Trinucleotides of breakpoint flanks more likely occurred in nCC/CCn on mtDNA genome and less likely in nTT/TT on both nuclear and mtDNA genomes, particularly for ultra-rare NUMTs. The same trend was not seen in the tumour-specific NUMTs (**b**), indicating the signal is driven by biology, but not the sequencing artefacts. **b**. Frequencies of trinucleotides around tumour-specific NUMTs breakpoints in the nuclear genome (top) and mtDNA genomes (bottom), tumour-specific NUMTs and the expected frequencies shown in the different colours. P values # < 0.1, * < 0.05, < 0.01 **, < 0.001 ***, < 0.0001 **** (two-sided Fisher's exact test) (Supplementary Table 6). **c**. Distribution of the distance between PRDM9 binding sites and tumour-specific NUMTs within each tumour type. **d**. Age of NUMTs estimated in this study. Y axis shows the frequencies of NUMTs in African and non-African populations. The frequencies of NUMTs were different between African and non-African, particularly for the older NUMTs which were more common seen in African population.

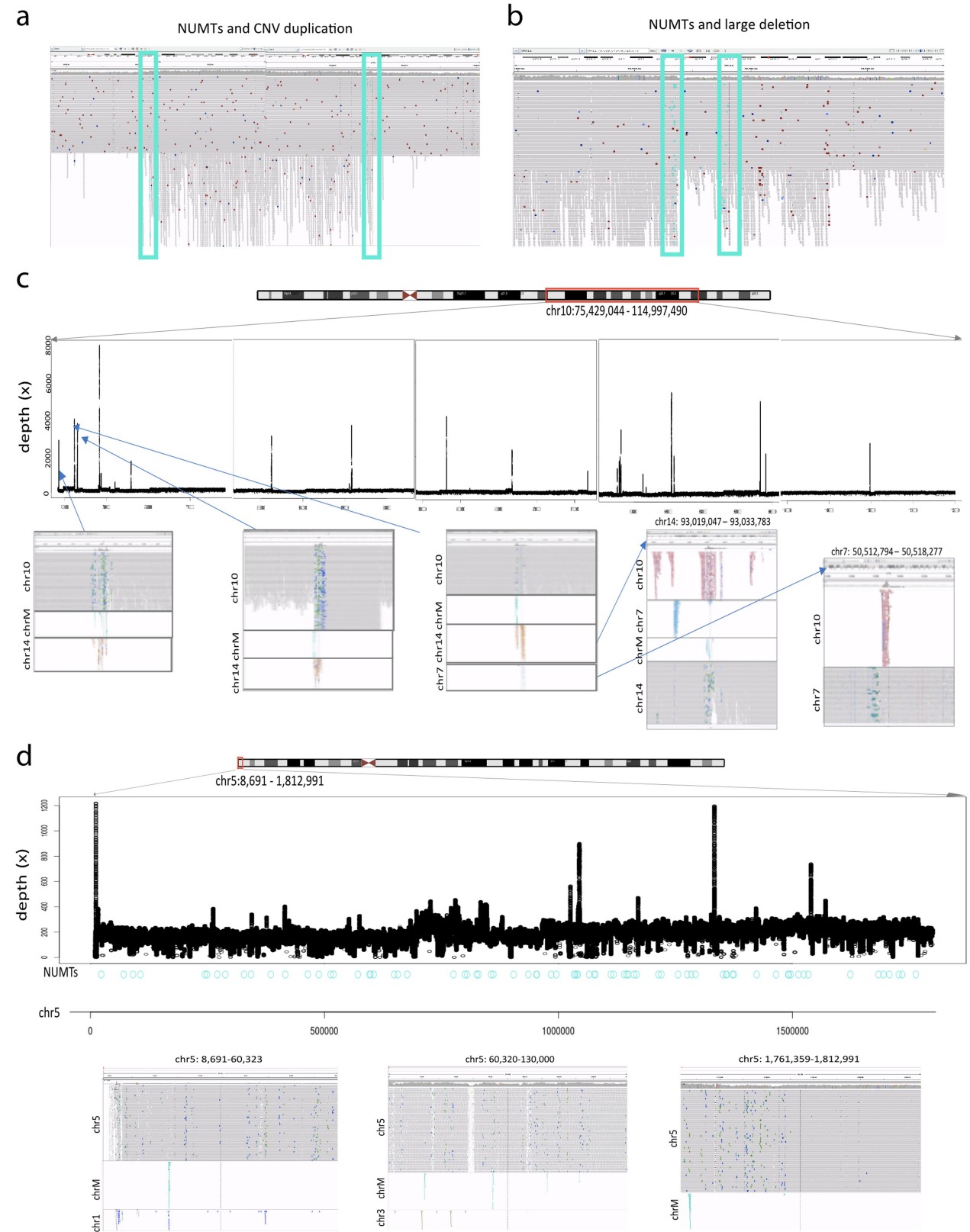

**Extended Data Fig. 9 | IGV alignments of NUMTs and nuclear chromosomal structure variations. a**. An example of mtDNA fragment inserted into two edges of a CNV duplication. **b**. An example of mtDNA fragment inserted into two edges of a large deletion. Teal bars indicate the aligned reads which mapped to the nuclear DNA where their mates mapped to the mtDNA, and were highlighted in the teal. **c. d**. Two examples of cancer genomes carrying mito-chromothripsis observed in this study. **c**. The sequencing depth of

nuclear genome is shown at the top panel. Examples of the read alignment of NUMTs from IGV are shown at the bottom. Reads are coloured by the pair orientation and the chromosome on which their mates can be found. **d**. The sequencing depth of nuclear genome is shown at the top panel. Teal dots are the locations of NUMT insertions. Examples of the read alignment of NUMTs from IGV are shown at the bottom. Reads are coloured by the pair orientation and the chromosome on which their mates can be found.

# Reporting Summary

## Statistics

For all statistical analyses, confirm that the following items are present in the figure legend, table legend, main text, or Methods section.

| n/a | Confirmed | |
|-----|-----------|---|
| ☐ | ☒ | The exact sample size (*n*) for each experimental group/condition, given as a discrete number and unit of measurement |
| ☐ | ☒ | A statement on whether measurements were taken from distinct samples or whether the same sample was measured repeatedly |
| ☐ | ☒ | The statistical test(s) used AND whether they are one- or two-sided *Only common tests should be described solely by name; describe more complex techniques in the Methods section.* |
| ☐ | ☒ | A description of all covariates tested |
| ☐ | ☒ | A description of any assumptions or corrections, such as tests of normality and adjustment for multiple comparisons |
| ☐ | ☒ | A full description of the statistical parameters including central tendency (e.g. means) or other basic estimates (e.g. regression coefficient) AND variation (e.g. standard deviation) or associated estimates of uncertainty (e.g. confidence intervals) |
| ☐ | ☒ | For null hypothesis testing, the test statistic (e.g. *F*, *t*, *r*) with confidence intervals, effect sizes, degrees of freedom and *P* value noted *Give P values as exact values whenever suitable.* |
| ☒ | ☐ | For Bayesian analysis, information on the choice of priors and Markov chain Monte Carlo settings |
| ☒ | ☐ | For hierarchical and complex designs, identification of the appropriate level for tests and full reporting of outcomes |
| ☐ | ☒ | Estimates of effect sizes (e.g. Cohen's *d*, Pearson's *r*), indicating how they were calculated |

*Our web collection on statistics for biologists contains articles on many of the points above.*

## Software and code

Policy information about availability of computer code

| Data collection | The sequence data included in this study is available in Genomics England Research Environment. |
|---|---|
| Data analysis | ISAAC (viSAAC-03.16.02.19); Strelka (v2.4.7); Manta (v0.28.0); Canvas (v1.3.1); PLINK(v1.9); BCFTools (v1.3.1); SAMtools (v1.9); MToolBox (v1.0); VarScan2; HaploGrep2; BLAT; bedtools (v2.19.1); R (v.3.6 to v.4.0); minimap2(v2.17); Nanoplot (v1.26.0); Nanopolish (v0.13.3); Python3; Circos (v0.69); IGV (v2.5); Shiny (v1.7.1).VerifyBamID (v1.1.3); ConPair (v0.2); R Package UMAP (v0.2.7.0); R Package M3C (v1.18); samblaster (v0.1.25); blat (v3.5); bedtools (v2.19.1); CAP3; BioPython (v1.77); Matplotlib(v3.3.1) Custom code used in the study is available at: https://github.com/WeiWei060512/NUMTs-detection.git. The software and methods used to do the analysis are all cited and described in the manuscript. |

For manuscripts utilizing custom algorithms or software that are central to the research but not yet described in published literature, software must be made available to editors and reviewers. We strongly encourage code deposition in a community repository (e.g. GitHub). See the Nature Portfolio guidelines for submitting code & software for further information.

## Data

Policy information about availability of data

All manuscripts must include a data availability statement. This statement should provide the following information, where applicable:
- Accession codes, unique identifiers, or web links for publicly available datasets
- A description of any restrictions on data availability
- For clinical datasets or third party data, please ensure that the statement adheres to our policy

The sequence data is available for analysis through the Genomics England data warehouse https://www.genomicsengland.co.uk/understanding-genomics/data/; Homo Sapiens NCBI GRCh38 assembly can be found at https://www.ncbi.nlm.nih.gov/assembly/;

GENCODE v29 can be found at https://www.gencodegenes.org/human/release_29.html;
Human genome annotation files can be found at https://hgdownload.soe.ucsc.edu/goldenPath/hg38/database/;
The ancestral mitochondrial sequences from Chimpanzee can be found at https://www.ensembl.org/Pan_troglodytes/Info/Index;

# Field-specific reporting

Please select the one below that is the best fit for your research. If you are not sure, read the appropriate sections before making your selection.

☒ Life sciences  ☐ Behavioural & social sciences  ☐ Ecological, evolutionary & environmental sciences

For a reference copy of the document with all sections, see nature.com/documents/nr-reporting-summary-flat.pdf

# Life sciences study design

All studies must disclose on these points even when the disclosure is negative.

| | |
|---|---|
| Sample size | We studied 68,348 genomes in Genomics England Rare Disease Project and 26,488 cancer genomes from Genomics England Cancer Project. After all quality control steps, we included 53,574 genomes in Rare Disease Project and 12,509 tumour-normal tissue pairs in Cancer Project. We analysed all of the available data at the time we started this study, and reported the results of our statistical analyses with confidence intervals. |
| Data exclusions | We excluded the genomes aligned to the Homo Sapiens NCBI hg19 assembly, and failed either whole genome sequencing QCs or mitochondrial genome QCs. The details are described in the manuscript - Methods. |
| Replication | Replication was not possible. We used all available data in our primary analysis. |
| Randomization | We performed an observational study on all available data. Randomisation was not appropriate because our study design did not involve experimental interventions. |
| Blinding | We performed an observational study. No experimental interventions were performed, so blinding was not necessary |

# Reporting for specific materials, systems and methods

We require information from authors about some types of materials, experimental systems and methods used in many studies. Here, indicate whether each material, system or method listed is relevant to your study. If you are not sure if a list item applies to your research, read the appropriate section before selecting a response.

## Materials & experimental systems

| n/a | Involved in the study |
|---|---|
| ☒ | ☐ Antibodies |
| ☒ | ☐ Eukaryotic cell lines |
| ☒ | ☐ Palaeontology and archaeology |
| ☒ | ☐ Animals and other organisms |
| ☐ | ☒ Human research participants |
| ☒ | ☐ Clinical data |
| ☒ | ☐ Dual use research of concern |

## Methods

| n/a | Involved in the study |
|---|---|
| ☒ | ☐ ChIP-seq |
| ☒ | ☐ Flow cytometry |
| ☒ | ☐ MRI-based neuroimaging |

## Human research participants

Policy information about studies involving human research participants

| | |
|---|---|
| Population characteristics | We studied 68,348 genomes in Genomics England Rare Disease Project and 26,488 cancer genomes from Genomics England Cancer Project. After all quality control steps, we included 25,436 male and 28,138 females aged from 0 to 99y in Rare Disease Project (Extended Data Fig.1a&b) and 12,509 tumour-normal tissue pairs from 21 different cancer types in Cancer Project (Extended Data Fig.6a&b). More information can be found in the website https://www.genomicsengland.co.uk/about-genomics-england/the-100000-genomes-project/. |
| Recruitment | The Genomics England 100,000 Genomes Rare Disease Project enrolled people with a high likelihood or clear evidence of a rare inherited disorder. More information can be found in the website https://www.genomicsengland.co.uk/about-genomics-england/the-100000-genomes-project/information-for-gmc-staff/rare-disease-documents/rare-disease-eligibility-criteria/. Genomics England Cancer Project recruited patients with conditions corresponding to the eligibility criteria. The details can be found in the website https://www.genomicsengland.co.uk/about-genomics-england/the-100000-genomes-project/information-for-gmc-staff/cancer-programme/eligibility/. |
| Ethics oversight | Ethical approval was provided by the East of England Cambridge South national research ethics committee under reference |

Ethics oversight

number: 13/EE/0325, with participants providing written informed consent for this approved study. All consenting participants in the Rare Disease arm of the 100,000 Genomes Project were enrolled via thirteen centres in the National Health Service covering all NHS patients in England.

Note that full information on the approval of the study protocol must also be provided in the manuscript.

