## [Peer Review File · Nature]

Manuscript Title: Nuclear-embedded mitochondrial DNA sequences in 66,083 human genomes

Reviewer Comments & Author Rebuttals

Reviewer Reports on the Initial Version:

Referees' comments:

Referee #1 (Remarks to the Author):

In this manuscript, Wei and colleagues describe the molecular landscape of NUMTs in both the germline and somatic tumor genomes. They address a number of overarching open questions related to the genesis of NUMTs as well as their prevalence, and as the authors themselves state, it will likely prove valuable for clinical applications, such as the identification of disease-causing alleles in mtDNA. While certainly a significant body of work for the mitochondrial genetics field, I did not find a conceptual innovation or biological discovery that reshaped my understanding of the biology of NUMTs. There have been prior descriptions of somatic transfer of mtDNA in cancer genomes (e.g. from Ju et al in *Genome Research* 2015) as well as their de novo generation (see the review by Puertas and Gonzalez-Sanchez, *Genome* 2020).

While I did not find the work revolutionary, I did on the other hand find it comprehensive and extensively hypothesis-generating. My impression is that in the context of a character-count-limited manuscript, the authors were forced to be brief where I wanted to read more. These impressions, as well as other questions, suggestions, and concerns, are described below.

1. In the beginning of the manuscript, the authors comment that misinterpretation of NUMTs as bona fide mtDNA would potentially lead to erroneous clinical diagnoses and interpretations of mtDNA inheritance. I was curious to know if any of the NUMTs described throughout the manuscript are examples of this. Presumably such NUMTs would satisfy two criteria: they contain a disease-causing or disease-associated variant, and the NUMT (through duplication or other amplification) is at sufficiently high copy number so as to appear at a high enough heteroplasmy to cause disease.
2. Presumably NUMTs are under largely neutral selection once translocated to the nuclear genome. Do you see evidence of this? Is there any evidence that some NUMTs, or alternatively some circumscribed loci in the mtDNA which have migrated to the nucleus, remain under purifying selection? This is an open-ended question and a difficult one, but I am grasping for a sense of whether NUMTs play any functional role once embedded in nuclear DNA.
3. Do the authors have a hypothesis for why some chromosomes appear to be the targets of NUMT transfer more than others? Chromosomal length does not appear to be the culprit.
4. I found the tumor analysis quite interesting. Presumably (please correct me if mistaken) much of the "normal" data from the Genomics England Cancer project came from matched blood from cancer patients, as is typical of cancer sequencing projects. While blood is often a perfectly sufficient surrogate "normal" tissue in many cases, I think here it raises numerous interesting (and I mean this literally, not troubling, just interesting) questions: first, is there a difference in the selective pressure for/against NUMT-genesis in the blood lineage vs other somatic lineages, including tumor lineages? In the TCGA (exome sequencing), there are a subset of matched normal tissue samples, e.g. normal colon from colorectal cancer patients. If these are available from Genomics England, how do the rates of NUMT-genesis compare in tumor, normal tissue, and

normal blood?

5. Related to the above question, the authors report that the rate of NUMT-genesis is 100-fold higher in tumors than (blood) normals. 100-fold is remarkable. Is that a result of increased sequencing depth? If not, do the authors think that, potentially, it is the result of relaxed constraints on genomic diversification in tumors? For example, some processes such as homologous recombination deficiency can promote enormous and ongoing genomic diversity, even at the level of individual cells.

6. I had trouble interpreting the first few statements in the "Adverse consequences" section. Is it the case that only ultra-rare NUMTs were found in coding DNA sequences? Does that imply that NUMTs in coding sequences are recent events? Under purifying selection?

7. Are the authors indicating that the FUS:DDIT3 fusion was caused by a NUMT insertion? Is there evidence for this in the data, e.g. was this an ultra-rare NUMT? Or, alternatively, were they simply coincidentally near the same locus?

8. A very general comment: I found that the authors very briefly described many key points from the Supplementary Data. I suspect this was simply the outcome of trying to keep the manuscript succinct, and I don't tally this as a point against them. But, should such length constraints become relaxed, I would encourage the authors to elaborate more. At moments in the manuscript, especially in the introduction, their prose was so pleasant, lucid, and illuminating, and I would have liked to read more of it.

Referee #2 (Remarks to the Author):

In the manuscript entitled "Characterization of nuclear-embedded mitochondrial DNA sequences in 66,083 human genomes", Wei and colleagues conducted a large population-scale genome study for understanding Nuclear Mitochondrial DNA segment, or NUMTs. Although similar studies have been carried out before (i.e., for germline, Dayama et al., NAR 2014 (cited in the manuscript); for cancers, Ju et al., Genome Research 2015 (cited in the manuscript) and Yuan et al., Nature Genetics 2020 (not cited in the manuscript)), the beauty of this work is stemming from its sample size, at least ten times as many sequences. Given that NUMTs are rare events compared to single nucleotide variants (SNVs), the size of samples is essential to reveal the overall landscape of NUMTs in human genomes. Using statistical power from the largest cohort of whole-genome sequences, the authors detected >1K of NUMT sites, most of which are previously undiscovered, in both germline and somatic (tumors) human tissues. Using the extensive catalog of NUMTs, the authors have the power to explore many factors that are associated with NUMTs, which are previously challenging to conduct: the authors identified the size distribution of NUMT inserts, sequence contexts which predispose to NUMT insertion, functional contexts of NUMT insertions, to name a few.

Technically, the analyses are well-conducted using standard bioinformatics algorithms. Combining oxford nanopore platform, the authors also characterized the entire structure of a few NUMT segments as well as DNA methylation.

Overall, these are novel and interesting observations that are helpful to understand forces that are modifying human genome sequences. The manuscript reads well, and analyses/interpretations are straightforward. Here I provide some suggestions that may be necessary to improve the manuscript.

1. As the authors briefly mentioned, the genomic landscape of NUMTs may be different according to ethnic groups. Because the authors used Genomic England genome datasets, most samples in the cohort would be Europeans. It will be helpful if the ethnic composition of the cohort is shown

explicitly in a panel of the main figure rather than in the Methods section or by Supp Figures.

2. The list of polymorphic NUMTs ($n=1,615$ likely) identified in the study will be a great resource as a "dictionary" for future studies. Therefore, their genomic positions, characteristics, population allele frequency, and samples that have each specific NUMT should be shared by the scientific community explicitly by a Supplementary Data Table.

3. In the human reference genome, there are already ~ 600 NUMTs. I am wondering all the 1,615 NUMTs identified in this study are non-referenced.

4. The authors claim they identified three de novo germline NUMTs, which directly calculates the mutation rate per generation. The analysis implies that the origin of the insertion would be mitochondria. On the contrary, there is a possibility that the origin of the segment is other NUMTs, rather than mitochondria. For example, by MMBIR or other mechanisms, mitochondria-like sequences in the nuclear genome can be sources for de novo insertion. Are the authors able to rule out the possibility? In Ju et al., Genome Research 2014, the authors used mtDNA polymorphisms to differentiate the origin.

4-1. In line with the comment above, short NUMTs (24bp...) are difficult to pinpoint their origin.

5. Lines 127-128. The copy number of Chromosome X is half in males; thus the chromosome is less explored. Is it considered in the analysis?

6. Atlas of tumor-specific NUMTs. Yuan Y and colleagues recently analyzed NUMTs in $\sim 2,500$ cancers (Nature Genetics 2020). It should be cited here, and the authors may want to make some comparisons with the previous analysis.

7. Line 146. The authors identified 6.5 NUMTs in cancer genomes, ~ 2 more than its matched germline. Given the rare frequency in somatic lineages, most of them should be germline ones, inherited from the parent genome. In this scenario, cancer should have the same number of NUMTs compared to its germline genome. In reality, due to many loss-of-heterozygosity (copy number loss), we expect to find a lower number of germline NUMTs in the cancer genome. Why do the authors find more NUMTs in the cancer genome?

8. Description of somatic and germline NUMTs in cancers should be separately described.

9. NUMTs in the tumor are associated with APOBEC-mediated point mutations (SBSs 2 and 13). It can be interesting because APOBEC can induce double-strand DNA breaks - which facilitate insertion of mtDNA.

10. Line 195. FHIT gene is also interesting because the locus is the fragile site of the genome.

11. Line 223. What is the 'per mutation test'?

12. Fig 4i. These are fascinating cases. Are the NUMTs distributed in multiple chromosomes as described in the main manuscript? In Fig 4i, they are localized in chromosome 10 and chromosome 5 in each genome. Supp Fig 22 is not very clear.

12-1. Fig 4i. What is the allele frequency of these NUMT events? Is there any possibility of sequencing artifacts?

13. Please clarify some sentences or provide references.

A. Lines 202-204

B. Lines 241. ... NUMT insertion events through recombination.

Referee #3 (Remarks to the Author):

In the application from Wei et al., the authors characterized 66000 NUMTS and showed that NUMTs arise every $1/10^4$ births and in $1/10^3$ cancers. NUMTS have been considered a historical marker of mitochondrial DNA (mtDNA) escape, a carryover from mtDNA reduction, and marking hundreds to thousands of generations. With the new pipeline deployed by the authors, NUMTs move into evolutionary relevance on a single generational level. It may also be a cause of some cancers. This study is highly significant and of interest to oncology, human genetics, mitochondrial disorders, mobile genetic elements (as the data show NUMTs move!) and evolutionary biologists.

An argument can be made that this manuscript should be in Nature Genetics as it relates to genetic alterations and disease origins. That would be a good outcome for this study. However, it isn't every day that dogma around a genetic element, which is really what a NUMT is, changes from a multigenerational scale marker to a single generational event with disease implications. Multiple scales of selection are occurring with NUMTS and will potentially create a newly expanded genetic and evolutionary interest area. This manuscript redefining this genetic element might warrant publication in Nature.

Comments

- Intro and summary are almost identical and could perhaps differentiate further
- A searchable database should be considered if the breakpoint locations are that important for the world to know.
- Differences meant to be communicated by the bar chart in Fig1e are not clear.
- Figure 2 –Figure 2c lacks sufficient description – is the first panel for the whole data set? Onward, it is overly dense for African, East Asian, and American. It seems like there has to be another way to convey this high-density data. The line width must be too large. So many overlapping colours that locations have turned black and no longer communicate frequency. This panel is more of a distribution indication than a frequency indication (it might be out of calibration because of line width).
- One could argue that panels 2e, f are not needed in the main figures. I would just put this in extended.
- It is unclear how panel 2i supports no association with mtDNA deletion breakpoints as there appears to be breakpoint enrichment in the D-loop and more dloop sequence in numts. How was this determined statistically? Again, not enough is said about these panels to make them particularly useful or informative.
- Page 7, lines 199-202. "One myxoid liposarcoma tumour had a FUS:DDIT3 chimeric fusion oncoprotein found in 90% of myxoid liposarcomas (Fig. 3j, Extended Data Fig.17), caused by a complex rearrangement with NUMT insertion, indicating a causal role." The sentence construction makes it unclear whether the NUMT insertion or the complex rearrangement of the NUMT insertion, causes 90% of myxoid liposarcomas or just one tumor. Please clarify.
- Pg 7, line 211-214, it is unclear what exactly is meant by "implicating mtDNA repeat sequences in NUMT insertion," as 2-4 bp of poly-C track isn't what is typically called a mtDNA repeat (more like 11 nt). If this is just poly-C or micro-homology, then limit the word choice to one of those two concepts.
- PRDM9-mediated effects seem like a hypothesis arising from the data rather than a supportable conclusion. The discussion of PRDM9 appears to interrupt the discussion of poly-C and microhomology domains driving either DSB repair or recombination that bracket the mentioning of that protein.
- Avoid dangling articles such as this and those!

Author Rebuttals to Initial Comments:

Nature 2022-01-00173: response to referees' comments

Referee #1:

Referee: In this manuscript, Wei and colleagues describe the molecular landscape of NUMTs in both the germline and somatic tumor genomes. They address a number of overarching open questions related to the genesis of NUMTs as well as their prevalence, and as the authors themselves state, it will likely prove valuable for clinical applications, such as the identification of disease-causing alleles in mtDNA. While certainly a significant body of work for the mitochondrial genetics field, I did not find a conceptual innovation or biological discovery that reshaped my understanding of the biology of NUMTs. There have been prior descriptions of somatic transfer of mtDNA in cancer genomes (e.g. from Ju et al in *Genome Research* 2015) as well as their de novo generation (see the review by Puertas and Gonzalez-Sanchez, *Genome* 2020).

Response: Previous reports of apparently *de novo* NUMTs were *inferred* by studying unrelated individuals (as reviewed by Puertas and Gonzalez-Sanchez 2020). One of our key discoveries was to directly show that NUMTs arise during germ line transmission in humans over a single generation. This has not been done before because the events are rare (1 in every ~4000 births). This required a dataset >20-fold larger than either of the two studies cited above. This is important because it allowed us to directly measure the rate of NUMT insertion during germline transmission for the first time, compare this rate to the somatic transfer in cancers, and show no evidence of a parent-of-origin effect, as previously speculated (and discussed in Puertas and Gonzalez-Sanchez 2020).

Referee: While I did not find the work revolutionary, I did on the other hand find it comprehensive and extensively hypothesis-generating. My impression is that in the context of a character-count-limited manuscript, the authors were forced to be brief where I wanted to read more. These impressions, as well as other questions, suggestions, and concerns, are described below.

Response: The reviewer is absolutely correct. Our analysis of a very rich novel dataset has validated some previous findings and raised many new hypothesis – but also led to totally novel discoveries. We have expanded the novel findings in the revised manuscript, focussing on NUMT sequence evolution, incorporating a new main figure, and Extended Data figures and tables.

Referee: 1. In the beginning of the manuscript, the authors comment that misinterpretation of NUMTs as bona fide mtDNA would potentially lead to erroneous clinical diagnoses and interpretations of mtDNA inheritance. I was curious to know if any of the NUMTs described throughout the manuscript are examples of this. Presumably such NUMTs would satisfy two criteria: they contain a disease-causing or disease-associated variant, and the NUMT (through duplication or other amplification) is at sufficiently high copy number so as to appear at a high enough heteroplasmy to cause disease.

Response: To address the reviewer's question we have looked for 89 recently curated and confirmed pathogenic mtDNA mutations (Ref ¹) in our data. To achieve this, we re-constructed the genomic sequences of 931 different NUMTs. Six mutations were identified in the NUMT sequences that corresponded to known pathogenic mtDNA mutations. However, long read sequencing showed that these were only present in single copy NUMTs. Thus, it is highly unlikely that these would lead to 'pseudo-heteroplasmies' at >5% levels, leading to false positive diagnoses.

We have included this new information in the revised manuscript as follows:

P10.

Six of the 5637 NUMT-specific variants corresponded to known pathogenic mtDNA mutations in humans¹: 8993G, 12706C, 13042A, 13051A, 13094C and 14849C, listed according to the original mtDNA site. Long-read sequencing showed that all were within single copy NUMTs, making them unlikely to cause high levels of pseudo-heteroplasmy leading to a false diagnosis of mtDNA disease².

Referee: 2. Presumably NUMTs are under largely neutral selection once translocated to the nuclear genome. Do you see evidence of this? Is there any evidence that some NUMTs, or alternatively some circumscribed loci in the mtDNA which have migrated to the nucleus, remain under purifying selection? This is an open-ended question and a difficult one, but I am grasping for a sense of whether NUMTs play any functional role once embedded in nuclear DNA.

Response: We have looked for evidence of purifying selection in the NUMT sequences by calculating ratio of non-synonymous to synonymous variants (Ka/Ks) within embedded NUMTs, and have compared this to true mtDNA. This analysis shows that once the mtDNA has translocated into the nucleus, it is no longer under evolutionary constraint.

We have included these new findings in the revised manuscript as follows:

P10.

The Ka/Ks of the NUMT-specific variants was greater than real mtDNA (Total group A = 3.2, vs mtDNA $P = 0$, odds ratio=6.6, 95% CI = 6.0-7.2; Subgroup B = 3.5, vs. mtDNA $P = 0$, odds ratio=7.1, 95% CI = 6.4-8.0; Subgroup C: 2.1 vs. real mtDNA $P = 1.73 \times 10^{-6}$, odds ratio=4.2, 95%CI = 2.2-8.2; Fisher Exact Test) (Fig. 6a). Moreover, Ka/Ks of Subgroup C was no different to the null hypothesis for random mutations of 2.91 ($P = 0.25$, odds ratio=1.20, 95%CI = 0.72-2.64, Fisher Exact Test) (Fig. 6a). Therefore, once translocated to the nucleus, the same genomic sequence is no longer under evolutionary constraint, consistent with NUMTs having no functional role.

Referee: 3. Do the authors have a hypothesis for why some chromosomes appear to be the targets of NUMT transfer more than others? Chromosomal length does not appear to be the culprit.

Response: We have now carried out this analysis, which shows a correlation between chromosome length and the number of NUMTs. Thus, chromosome length does appear to be a contributory factor in determining the number of NUMTs per chromosome.

We have included these new findings in the revised manuscript as follows:

P5.

Overall, we observed a strong positive correlation between the length of each chromosome and the number of NUMTs detected on each chromosome after accounting for other genomic features ($P = 1.42 \times 10^{-6}$, linear regression test).

Referee: 4. I found the tumor analysis quite interesting. Presumably (please correct me if mistaken) much of the “normal” data from the Genomics England Cancer project came from matched blood from cancer patients, as is typical of cancer sequencing projects. While blood is often a perfectly sufficient surrogate “normal” tissue in many cases, I think here it raises numerous interesting (and I mean this literally, not troubling, just interesting) questions: first, is there a difference in the selective pressure for/against NUMT-genesis in the blood lineage vs other somatic lineages, including tumor lineages? In the TCGA (exome sequencing), there are a subset of matched normal tissue samples, e.g. normal colon from colorectal cancer patients. If these are available from Genomics England, how do the rates of NUMT-genesis compare in tumor, normal tissue, and normal blood?

Response: None of the tumour samples had matched WGS from more than one other tissue, although 344 were from saliva and 33 were from fibroblasts. The average number of

germ-line NUMTs was no different between saliva, fibroblast and blood samples (average detected NUMT was 4.7 in saliva, 5.0 in fibroblasts and 4.9 in blood samples, saliva vs blood $P = 0.24$, estimate = -0.1, fibroblast vs blood $P = 0.67$, estimate = -0.1, linear regression test). Cancer germ-line NUMTs were also no different to the germline NUMTs defined in the analysis rare disease families. Thus, we found no evidence that there are more NUMTs in the germline DNA from cancer patients.

We have included these new findings in the revised manuscript, along with new Extended Data figures:

P6.

Overall, tumours had a greater mean number of NUMTs (6.5, s.d. 2.2) than their normal tissue (4.8, s.d. 1.6, $P < 2.2 \times 10^{-16}$, Wilcoxon rank sum test) (Fig.4a, Extended Data Fig.20). This difference likely reflects the tumour itself, rather than the normal tissue in each case, because the mean number of NUMTs did not differ between different normal tissue types (average detected NUMT was 4.7 in saliva cells, 5 in skin fibroblasts, and 4.9 in blood samples, saliva vs blood $P = 0.24$, estimate = -0.1, fibroblast vs blood $P = 0.67$, estimate = -0.1, linear regression test)(Extended Data Fig.21). The frequency of cancer germ-line NUMTs was no different to the frequency of germline NUMTs measured in the Rare Disease Project participants ($P = 0.924$, linear regression test accounting for sequencing depth. Extended Data Fig. 22).

Referee: 5. Related to the above question, the authors report that the rate of NUMT-genesis is 100-fold higher in tumors than (blood) normals. 100-fold is remarkable. Is that a result of increased sequencing depth? If not, do the authors think that, potentially, it is the result of relaxed constraints on genomic diversification in tumors? For example, some processes such as homologous recombination deficiency can promote enormous and ongoing genomic diversity, even at the level of individual cells.

Response: As shown in Extended Data Fig. 33, the sequencing depth in tumours was greater than the germ line. As shown in Extended Data Fig. 34 and mentioned in the main text, there was a weak correlation between the mtDNA sequencing depth and the number of NUMTs detected. However, with an R^2 of 0.134, it is highly unlikely that the difference in coverage accounts for the difference in NUMT genesis seen in tumours. We agree with the referee that relaxed constraints on genomic diversification in tumours and increased genomic instability in tumours contributes to the generation of NUMTs in tumours. We have expanded our discussion of this as follows:

P7.

Taken together, these findings suggest that a combination of local sequence characteristics, genome instability, and less opportunity for selection to remove specific NUMTs due to relaxed evolutionary constraints, explains why the NUMT landscape differs from the germline.

Referee: 6. I had trouble interpreting the first few statements in the “Adverse consequences” section. Is it the case that only ultra-rare NUMTs were found in coding DNA sequences? Does that imply that NUMTs in coding sequences are recent events? Under purifying selection?

Response: The referee is correct, we only found ultra-rare NUMTs within coding DNA sequences, and we agree that one explanation is that purifying selection has removed NUMT-events disrupting essential gene function. We alluded to this in the original manuscript (P7), and have expanded this section to place greater emphasis on this point, as follows:

P8.

*No common or rare NUMTs ($F > 0.1\%$) were found in the coding DNA sequences (CDS, $P = 0.039$ per mutation test), and none were predicted to cause rare disease (See **Methods & Extended Data Results**), consistent with NUMTs being under evolutionary constraint.*

Referee: 7. Are the authors indicating that the FUS:DDIT3 fusion was caused by a NUMT insertion? Is there evidence for this in the data, e.g. was this an ultra-rare NUMT? Or, alternatively, were they simply coincidentally near the same locus?

Response: The FUS:DDIT3 fusion was due to a complex NUMT rearrangement shown in Fig. 4j and Extended Data Fig. 25. This was included in the original manuscript:

P8.

One myxoid liposarcoma tumour had a FUS:DDIT3 chimeric fusion oncoprotein found in 90% of myxoid liposarcomas³ (Fig. 4j, Extended Data Fig.25), caused by a complex rearrangement with an ultra-rare NUMT insertion, indicating a causal role.

Referee: 8. A very general comment: I found that the authors very briefly described many key points from the Supplementary Data. I suspect this was simply the outcome of trying to keep the manuscript succinct, and I don't tally this as a point against them. But, should such length constraints become relaxed, I would encourage the authors to elaborate more. At

moments in the manuscript, especially in the introduction, their prose was so pleasant, lucid, and illuminating, and I would have liked to read more of it.

Response: We are pleased to expand the text by incorporating new analyses. We have included a new section on the molecular evolution of NUMT sequences (P9/10) incorporating a new main figure (Fig. 6), 4 new Extended data figures, and a new Extended data table.

Referee #2:

Referee: In the manuscript entitled "Characterization of nuclear-embedded mitochondrial DNA sequences in 66,083 human genomes", Wei and colleagues conducted a large population-scale genome study for understanding Nuclear Mitochondrial DNA segment, or NUMTs. Although similar studies have been carried out before (i.e., for germline, Dayama et al., NAR 2014 (cited in the manuscript); for cancers, Ju et al., Genome Research 2015 (cited in the manuscript) and Yuan et al., Nature Genetics 2020 (not cited in the manuscript)), the beauty of this work is stemming from its sample size, at least ten times as many sequences. Given that NUMTs are rare events compared to single nucleotide variants (SNVs), the size of samples is essential to reveal the overall landscape of NUMTs in human genomes. Using statistical power from the largest cohort of whole-genome sequences, the authors detected >1K of NUMT sites, most of which are previously undiscovered, in both germline and somatic (tumors) human tissues. Using the extensive catalog of NUMTs, the authors have the power to explore many factors that are associated with NUMTs, which are previously challenging to conduct: the authors identified the size distribution of NUMT inserts, sequence contexts which predispose to NUMT insertion, functional contexts of NUMT insertions, to name a few.

Response: We thank the referee for the supportive comments. The largest previously published analysis of germ line NUMTs from the 1000 genomes dataset had ~50-fold less individuals than we studied and had ~10-fold less sequencing depth limiting NUMT detection. Crucially, we studied 8201 trios/duos allowing the first direct investigation of germ line NUMT transmission. We thank the referee for reminding us to cite Yuan et al., Nature Genetics 2020. We have added this reference to the revised manuscript (new Ref 26).

Referee: Technically, the analyses are well-conducted using standard bioinformatics algorithms. Combining oxford nanopore platform, the authors also characterized the entire structure of a few NUMT segments as well as DNA methylation.

Overall, these are novel and interesting observations that are helpful to understand forces that are modifying human genome sequences. The manuscript reads well, and analyses/interpretations are straightforward. Here I provide some suggestions that may be necessary to improve the manuscript.

As the authors briefly mentioned, the genomic landscape of NUMTs may be different according to ethnic groups. Because the authors used Genomic England genome datasets, most samples in the cohort would be Europeans. It will be helpful if the ethnic composition of the cohort is shown explicitly in a panel of the main figure rather than in the Methods section or by Supp Figures.

Response: We have moved the two figure panels describing the ethnic groups (original Extended Data Figs. 4a & b) into the main text as Fig. 2a.

Referee: The list of polymorphic NUMTs ($n=1,615$ likely) identified in the study will be a great resource as a "dictionary" for future studies. Therefore, their genomic positions, characteristics, population allele frequency, and samples that have each specific NUMT should be shared by the scientific community explicitly by a Supplementary Data Table.

Response: All of the NUMTs identified in this study are included in the Extended Data Table 1 & 2, and in an on-line searchable database <https://wei.shinyapps.io/numts/>

Referee: In the human reference genome, there are already ~600 NUMTs. I am wondering all the 1,615 NUMTs identified in this study are non-referenced.

Response: The 1615 refers to the rare/ultra-rare NUMTs we identified. We realise that this was not totally clear in the original manuscript, so have added the following summary statement to clarify the overall result:

P4.

Thus, combining the rare/ultra-rare and common NUMT data, we identified 1564 novel NUMTs not previously reported (Extended Data Table 1).

Referee: The authors claim they identified three de novo germline NUMTs, which directly calculates the mutation rate per generation. The analysis implies that the origin of the insertion would be mitochondria. On the contrary, there is a possibility that the origin of the segment is other NUMTs, rather than mitochondria. For example, by MMBIR or other mechanisms, mitochondria-like sequences in the nuclear genome can be sources for de novo insertion. Are the authors able to rule out the possibility? In Ju et al., Genome Research 2014, the authors used mtDNA polymorphisms to differentiate the origin.

Response: We have now determined the NUMT sequence for the likely *de novo* NUMTs described in the original manuscript. Although there were no single nucleotide variants in these NUMTs, the NUMT sequences only aligned to a single site in the nuclear genome of each child. This adds weight to our conclusion that in these three instances the NUMTs are *de novo*.

We have added the following text to the revised manuscript explaining this new analysis, revising new Fig.3d and Extended Data Fig. 12:

P5.

In each case the NUMT sequence did not align to any other site in the nuclear genome of the child, making it unlikely that the NUMTs originated from within the nuclear DNA.

Referee: In line with the comment above, short NUMTs (24bp...) are difficult to pinpoint their origin.

Response: We agree with the referee, and have made this point in the text as follows:

P5.

The de novo NUMT frequency is likely to be an under-estimate because of the difficulty determining the origin of short NUMTs.

Referee: Lines 127-128. The copy number of Chromosome X is half in males; thus the chromosome is less explored. Is it considered in the analysis?

Response: We detected 228 NUMTs on Chromosome X. As mentioned in the original manuscript, the frequency per Mb was lower than the autosomes (P5). The reasons for the different number of NUMTs per chromosome remains unclear, although chromosomal length is a contributory factor. We addressed this in response to referee 1 by including the following analysis:

P5.

Overall, we observed a strong positive correlation between the length of each chromosome and the number of NUMTs detected on each chromosome after accounting for other genomic features ($P = 1.42 \times 10^{-6}$, linear regression test).

We have also added additional detail, explaining the Chromosome X results as follows:

P5.

228 NUMTs were observed on Chromosome X, with the expected ~2 fold more in females than males (151 of the 28,138 females, and 75 of the 25,426 males, Fisher exact test $P = 1.713 \times 10^{-5}$, odd ratio = 1.824, 95 CI 1.374-2.441).

Referee: Atlas of tumor-specific NUMTs. Yuan Y and colleagues recently analyzed NUMTs in ~2,500 cancers (Nature Genetics 2020). It should be cited here, and the authors may want to make some comparisons with the previous analysis.

Response: We are pleased to include this reference (New Ref 26, cited at three relevant points in the revised manuscript on P6/7). We have also included a new Extended Data Table 3 comparing our analysis to the previously published results. This highlights similarities, such as the high frequency of NUMTs in bladder cancer, which we directly refer to in the revised manuscript (P7), but also highlights some inconsistencies in the Yuan paper. For example, in the main text they state ‘we did not find any positive [ie de novo NUMT] cases in blood, kidney, esophagogastric, liver, prostate and colorectal cancers’, but show a frequency greater than zero in Fig. 4a. It is difficult to reconcile this inconsistency from their published results.

Referee: Line 146. The authors identified 6.5 NUMTs in cancer genomes, ~2 more than its matched germline. Given the rare frequency in somatic lineages, most of them should be germline ones, inherited from the parent genome. In this scenario, cancer should have the same number of NUMTs compared to its germline genome. In reality, due to many loss-of-heterozygosity (copy number loss), we expect to find a lower number of germline NUMTs in the cancer genome. Why do the authors find more NUMTs in the cancer genome?

Response: We defined cancer-specific NUMTs as being present only in the tumour and not in matched normal samples from the same person, nor any other germline samples in either rare disease or cancer genomes. Thus, none of the cancer-specific NUMTs were inherited from the parents. We agree with the reviewer that cancer genome instability is likely to contribute to the high rate of new NUMTs in cancers. We have expanded our discussion of this point as follows:

P7.

Taken together, these findings suggest that a combination of local sequence characteristics, genome instability, and less opportunity for selection to remove specific NUMTs due to relaxed evolutionary constraints, explains why the NUMT landscape differs from the germline.

Referee: Description of somatic and germline NUMTs in cancers should be separately described.

Response: The original manuscript separately considered the 379 *de novo* NUMTs detected in the tumours. We have now expanded this section to show that the germline NUMTs in cancer patients (measured in the matched normal tissues) were no different to the germ-line results from the earlier rare disease genomes analysis. To address this, we have included the following text:

P6.

Overall, tumours had a greater mean number of NUMTs (6.5, s.d. 2.2) than their normal tissue (4.8, s.d. 1.6, $P < 2.2 \times 10^{-16}$, Wilcoxon rank sum test) (Fig.4a, Extended Data Fig. 20). This difference likely reflects the tumour itself, rather than the normal tissue in each case, because the mean number of NUMTs did not differ between different normal tissue types (average detected NUMT was 4.7 in saliva cells, 5 in skin fibroblasts, and 4.9 in blood samples, saliva vs blood $P = 0.24$, estimate = -0.1, fibroblast vs blood $P = 0.67$, estimate = -0.1, linear regression test)(Extended Data Fig. 21). These frequencies were no different to the frequency of germline NUMTs measured in the Rare Disease Project participants ($P = 0.924$, linear regression test accounting for sequencing depth. Extended Data Fig. 22).

Referee: NUMTs in the tumor are associated with APOBEC-mediated point mutations (SBSs 2 and 13). It can be interesting because APOBEC can induce double-strand DNA breaks - which facilitate insertion of mtDNA.

Response: We agree with the referee, and have added this point to the discussion as follows:

P9.

Signatures 2 and 13 are also enriched for APOBEC-mediated point mutations, which can also induce double-strand DNA breaks⁴.

Referee: Line 195. FHIT gene is also interesting because the locus is the fragile site of the genome.

Response: We agree with the referee, and have added this point to the discussion as follows:

P8.

....two in *FHIT* which is a fragile genomic site⁵, one each in *CNNNA2*, *DDIT3*, *WIF1*, *BCL11B*, *KDM5A* and *AKT2*.

Referee: Line 223. What is the 'per mutation test'?

Response: Apologies – this should read as two words not three – 'permutation test'. In this case, we randomly permuted the data to generate a random distribution under the null hypothesis. The P-value indicates the probability of seeing the actual observed data by chance. We have corrected these typos throughout the manuscript.

Referee: Fig 4i. These are fascinating cases. Are the NUMTs distributed in multiple chromosomes as described in the main manuscript? In Fig 4i, they are localized in chromosome 10 and chromosome 5 in each genome. Supp Fig 22 is not very clear.

Response: The referee is correct, the NUMTs are distributed on multiple chromosomes. We have added two new figures to the original Fig. 4i (new Fig. 5i) to show this more clearly.

Referee: Fig 4i. What is the allele frequency of these NUMT events? Is there any possibility of sequencing artifacts?

Response: As described in the methods, we took a conservative approach to detect the NUMTs, including the requirement of 5 or more concordant split reads to define the NUMT breakpoints. We validated this approach by independently using long-read sequencing, which confirmed the short-read approach in >99% of the NUMTs analysed. It is therefore highly unlikely that the NUMTs we report here are sequencing artifacts.

P3.

*Long-read sequencing validated our NUMT calling pipeline in >99% cases (182 of 184 NUMTs from 39 individuals, **Fig.1a, Extended Data Table 1**) (Methods).*

Referee: Please clarify some sentences or provide references.

A. Lines 202-204

B. Lines 241. ... NUMT insertion events through recombination.

Response: We have reworded these sections as follows:

A. P8.

Three private NUMTs in non-tumour tissue were not found in the matched breast tumours, potentially influencing prognosis through the loss of DSG2⁶ and TCAM1P⁷

B. P11.

The co-location of NUMTs with PRDM9 binding sites would facilitate their removal in the germ line because PRDM9 determines sites of recombination hot-spots during meiosis.

Referee #3:

Referee: In the application from Wei et al., the authors characterized 66000 NUMTS and showed that NUMTs arise every $1/10^4$ births and in $1/10^3$ cancers. NUMTS have been considered a historical marker of mitochondrial DNA (mtDNA) escape, a carryover from mtDNA reduction, and marking hundreds to thousands of generations. With the new pipeline deployed by the authors, NUMTs move into evolutionary relevance on a single generational level. It may also be a cause of some cancers. This study is highly significant and of interest to oncology, human genetics, mitochondrial disorders, mobile genetic elements (as the data show NUMTs move!) and evolutionary biologists.

An argument can be made that this manuscript should be in Nature Genetics as it relates to genetic alterations and disease origins. That would be a good outcome for this study. However, it isn't every day that dogma around a genetic element, which is really what a NUMT is, changes from a multigenerational scale marker to a single generational event with disease implications. Multiple scales of selection are occurring with NUMTS and will

potentially create a newly expanded genetic and evolutionary interest area. This manuscript redefining this genetic element might warrant publication in Nature.

Intro and summary are almost identical and could perhaps differentiate further

Response: We have added additional results to both sections to differentiate them more clearly.

Referee: A searchable database should be considered if the breakpoint locations are that important for the world to know.

Response: We have generated an on-line searchable database as suggested. This is available through <https://wwei.shinyapps.io/numts/>. We realise that this link was only included in the Methods, so was easy to miss. We have added the link at the end of the introduction to make this explicit.

P3.

The results are available in a searchable on-line database <https://wwei.shinyapps.io/numts/>.

Referee: Differences meant to be communicated by the bar chart in Fig1e are not clear.

Response: The bar chart shows the frequency of individuals carrying common NUMTs, rare NUMTs, ultra-rare NUMTs and private NUMTs. We have expanded the figure legends to make this clear.

Fig 1 legend.

Bar charts show the frequency of individuals carrying common, rare, ultra-rare and private NUMTs (the latter NUMTs being seen only seen in one family). 99.87% individuals carried at least one common NUMTs ($F > 1\%$), 26.2% individuals carried at least one NUMT with $F < 1\%$, 14.2% individuals carried at least one NUMT with $F < 0.1\%$ and 3.6% individuals carried at least one private NUMT (only seen in one family).

Referee: Figure 2 –Figure 2c lacks sufficient description – is the first panel for the whole data set? Onward, it is overly dense for African, East Asian, and American. It seems like there

has to be another way to convey this high-density data. The line width must be too large. So many overlapping colours that locations have turned black and no longer communicate frequency. This panel is more of a distribution indication than a frequency indication (it might be out of calibration because of line width).

Response: We agree with the referee. To address this we have regenerated all of the chromosome maps. We have expanded the map for the full dataset as new larger panel in Fig 2d, and expanded all of the other chromosome maps as Extended Data Figs. 4-8. We have revised and expanded the figure legends to match these changes.

Referee: One could argue that panels 2e, f are not needed in the main figures. I would just put this in extended.

Response: The figures show NUMT methylation data in different family members carrying the same NUMT. To our mind this is an important novel finding, but we would be pleased to move the figure panels to the Extended Data if the Editor requires this (now in revised Fig. 3b, c).

Referee: It is unclear how panel 2i supports no association with mtDNA deletion breakpoints as there appears to be breakpoint enrichment in the D-loop and more dloop sequence in numts. How was this determined statistically? Again, not enough is said about these panels to make them particularly useful or informative.

Response: We thank the referee for prompting us to look at this issue in more detail. We have now carried out a statistical analysis correlating the frequency of breakpoints seen in NUMTs and mtDNA deletion breakpoints in each mtDNA region. The correlation for germline NUMTs reached marginal significance ($P=0.047$), but there was a much stronger positive correlation for the *de novo* cancer-specific NUMTs ($P=0.004$). Although this could be due to deletions contributing to NUMT formation, we suspect this correlation reflects the enrichment for both NUMT and deletion breakpoints in the non-coding D-loop, implicating other mechanisms. We have included this new analysis and the discussion of this point as follows:

P5.

There was a weak correlation between the germline NUMT mtDNA breakpoints and the location of known deletion breakpoints in mtDNA reaching marginal significance ($P = 0.047$, $R^2 = 0.24$, Pearson correlation)

P7.

The mtDNA segments forming de novo tumour NUMTs differed from the germ line (**Fig.3e**), being less likely to involve MT-CO3 ($P = 7.7 \times 10^{-3}$), MT-ND4 ($P = 3.1 \times 10^{-3}$), MT-ND4L ($P = 3.4 \times 10^{-3}$) and MT-ND5 ($P = 5.3 \times 10^{-3}$), but >2.5-fold more likely to likely involve the D-loop ($P = 3.36 \times 10^{-36}$), largely because of a ~4-fold over-representation of breakpoints in the termination associated sequence 2 (TAS2, $P = 1.03 \times 10^{-7}$, Fisher Exact Test) (**Extended Data Fig 13**), also reflected in the mtDNA fragments (D-loop, $P = 5.51 \times 10^{-30}$, odds ratio = 2.00, 95% CI 1.77 – 2.25, Fisher Exact Test) (**Fig.3e, 4c**). This could explain the observed correlation between de novo NUMT breakpoints and known mtDNA deletion breakpoints ($P = 0.004$, $R^2 = 0.44$, Pearson correlation, **Fig. 3f, Extended Data Fig 14**), which also tend to cluster around the D-loop at the 3' end⁸.

And

P12.

Although this raises the possibility that mtDNA deletions are involved in NUMT formation, a more compelling explanation involves mtDNA transcription and associated replication, which originates in the D-loop⁹.

Referee: Page 7, lines 199-202. "One myxoid liposarcoma tumour had a FUS:DDIT3 chimeric fusion oncoprotein found in 90% of myxoid liposarcomas (Fig. 3j, Extended Data Fig.17), caused by a complex rearrangement with NUMT insertion, indicating a causal role." The sentence construction makes it unclear whether the NUMT insertion or the complex rearrangement of the NUMT insertion, causes 90% of myxoid liposarcomas or just one tumor. Please clarify.

Response: We have re-written the text to make this clearer:

P8.

One myxoid liposarcoma tumour had a FUS:DDIT3 chimeric fusion oncoprotein which was caused by a complex rearrangement involving a NUMT insertion (**Fig. 4j, Extended Data Fig. 25**). FUS:DDIT3 fusions are 90% of myxoid liposarcomas³, implicating the NUMT in carcinogenesis in this individual.

Referee: Pg 7, line 211-214, it is unclear what exactly is meant by "implicating mtDNA repeat sequences in NUMT insertion," as 2-4 bp of poly-C track isn't what is typically called a

mtDNA repeat (more like 11 nt). If this is just poly-C or micro-homology, then limit the word choice to one of those two concepts.

Response: We agree with the referee and have revised the text, referring directly to micro-homology as suggested.

Referee: PRDM9-mediated effects seem like a hypothesis arising from the data rather than a supportable conclusion. The discussion of PRDM9 appears to interrupt the discussion of poly-C and microhomology domains driving either DSB repair or recombination that bracket the mentioning of that protein.

Response: We agree with the reviewer, and have re-ordered this section (P8, Mechanisms of NUMT insertion and post-insertion modification) to avoid interrupting the discussion of poly-C and microhomology.

Referee: Avoid dangling articles such as this and those!

Response: We have tried to avoid dangling articles. We would be pleased to remove any specific examples that have been missed.

References cited in response to reviewers

1. Ratnaike, T.E. *et al.* MitoPhen database: a human phenotype ontology-based approach to identify mitochondrial DNA diseases. *Nucleic Acids Res* **49**, 9686-9695 (2021).
2. Wei, W. *et al.* Nuclear-mitochondrial DNA segments resemble paternally inherited mitochondrial DNA in humans. *Nat Commun* **11**, 1740 (2020).
3. Goransson, M. *et al.* The myxoid liposarcoma FUS-DDIT3 fusion oncoprotein deregulates NF-kappaB target genes by interaction with NFKBIZ. *Oncogene* **28**, 270-278 (2009).
4. Seplyarskiy, V.B. *et al.* APOBEC-induced mutations in human cancers are strongly enriched on the lagging DNA strand during replication. *Genome Res* **26**, 174-182 (2016).
5. Matsuyama, A. *et al.* Fragile site orthologs FHIT/FRA3B and Fhit/Fra14A2: evolutionarily conserved but highly recombinogenic. *Proc Natl Acad Sci U S A* **100**, 14988-14993 (2003).
6. Qin, S. *et al.* DSG2 expression is correlated with poor prognosis and promotes early-stage cervical cancer. *Cancer Cell Int* **20**, 206 (2020).
7. Rao, X. *et al.* MicroRNA-221/222 confers breast cancer fulvestrant resistance by regulating multiple signaling pathways. *Oncogene* **30**, 1082-1097 (2011).

8. Samuels, D.C., Schon, E.A. & Chinnery, P.F. Two direct repeats cause most human mtDNA deletions. *Trends Genet* **20**, 393-398 (2004).
9. Gustafsson, C.M., Falkenberg, M. & Larsson, N.G. Maintenance and Expression of Mammalian Mitochondrial DNA. *Annu Rev Biochem* **85**, 133-160 (2016).

Reviewer Reports on the First Revision:

Referees' comments:

Referee #1 (Remarks to the Author):

In this manuscript by Wei and colleagues, the authors carry out a comprehensive and thorough investigation of the genesis of NUMTs. They have addressed each of my comments clearly and succinctly.

In particular, I appreciate the authors clarifying the conceptual innovation of this work, which is that NUMT-genesis occurs over the time scale of a single generation, rather than hundreds to thousands of generations. I appreciate from reading the comments of the other referees that this redefines our understanding of NUMTs themselves. However, I remain unconvinced that this renewed understanding of their origins has immediate translational implications. While these events MAY be the functional origin of a key driver event in cancer, or disease-causing variant in other contexts, it seems that they rarely are.

A minor note: I did try to access their NUMT database, but could not do so as it was password protected. I also was unable to find a pointer to the code associated with analyzing the data, which would likely be useful for the field.

Referee #2 (Remarks to the Author):

In the revised manuscript entitled "Characterization of nuclear-embedded mitochondrial DNA sequences in 66,083 human genomes", Wei and colleagues reports novel NUMTs in human germline and cancer cells. As Reviewer #1 also pointed out, it is not easy to find a conceptual innovation. However, the beauty of this study comes from its huge sample size (>60K whole-genomes). It provides the most comprehensive analysis so far, and thus it will be an excellent resource for future studies. Therefore, together with outcomes from the downstream analyses, the list of NUMT calls is one of this manuscript's essential components.

Although the authors revised the manuscript according to my suggestion, I still feel some parts are not very clear. Therefore, I request some more clarification before its publication in the journal.

(1-in line with my previous question) The NUMT calls are now listed in Extended Data Table 1. I appreciate the effort. However, some more information may be necessary. For example, only NUMT frequency is categorized by a few groups, such as common, rare, and ultra-rare. The actual frequency can be shown in the table - potentially in each ethnic group. Ideally, showing sample id harboring specific NUMTs will be helpful for future studies. If the data size is not too big (I believe so), this information should be shared through an extended data table, not by the authors' website as it would not be as stable as the archive of the journal. In addition, the authors' website (<https://wwi.shinyapps.io/numts/>) can be accessed after the email address is registered.

(2-in line with my previous question) Throughout this manuscript, the definition of 'identified NUMT' is confusing. The human reference genome sequence, mostly contributed by a European individual, already has ~700 NUMTs ("referenced NUMTs"; Simone D et al., BMC Genomics 12:517), the vast majority of which, but not all, should be universal (shared by all human individuals) or common (shared by many individuals) NUMTs. Therefore, most human individuals have hundreds NUMTs in the genome (many of them should be 'referenced'). In this sense, the NUMT number reported by the authors (4.7 NUMTs; page 3; Fig 1d) may cause a lot of confusion. My interpretation of the 1,615 NUMTs reported here is 'the NUMTs absent in the human reference genome' ("non-referenced NUMTs"). Otherwise, 4.7 should be too small. It should be explicitly described in the first section of the Results.

(2-a) The meaning of "non-referenced NUMTs" is not identical to "novel NUMTs", because some of the 1,615 non-referenced NUMTs should be already reported by previous papers (e.g., Dayama et al., Nucleic Acids Research).

(2-b) If referenced NUMTs are not included in the list of 1,615 NUMTs as I supposed, there should be another concern. Not all the referenced NUMTs are shared by all human individuals. In extreme cases, some referenced NUMTs should be ultra-rare or private to the reference genome. Some rare NUMTs may not be counted in this manuscript, if it is referenced. If this scenario is true, Fig 1d should not be very meaningful, because it must be affected by the reference genome sequences. For example, the individual whose genome was used for the human genome project, will have no NUMT in the genome in this scenario. The NUMT number per genome should be counted absolutely.

(2-c) Therefore, I believe that referenced but polymorphic NUMTs should also be included (or considered) in this manuscript if the authors wish to count the number of NUMTs per individual, as shown in Fig 1d. Otherwise, the authors should at least explicitly describe that what they count here is 'non-referenced NUMTs'. In the latter case, the list of NUMTs reported in the manuscript is biased, and the impact of this study as an encyclopedia of NUMTs may be a bit compromised.

(2-d) It is helpful if the authors can also show their concept (i.e., referenced or non-referenced) of NUMT in Fig 1a.

(3-in line with my previous question) The length distribution of NUMTs ranges from 24bp to the whole mt genome. My previous question was how we could make sure that the origin of the three de novo NUMT is mitochondria in the germline of the parents. There are hundreds of (referenced) NUMTs in the genome, and those may be the source of the insertions. If the inserted sequences have mtDNA polymorphisms found in parents' germline DNA, it could be supporting evidence. Unfortunately, the authors did not find such variants. Although the authors claim that the NUMT sequences did not align to any other site in the nuclear genome of the child (reference genome sequence?-how can we align reads in the nuclear genome of the child?) it could not be perfect supporting evidence. I am wondering about the mapping quality of the NUMT supporting reads. The origin of the NUMT sequences could be the parents' non-referenced NUMTs.

(3-a) Although the authors introduced a new sentence in the revised manuscript that claims "the de novo NUMT frequency is likely to be an under-estimate...", I believe that the rate can be "over-estimate" as the origin of the insertion could be 'nuclear DNA'.

(4-in line with my previous question) I still do not understand why the authors found two more NUMTs in the cancer genome (6.5 v.s. 4.8 in cancers and matched normals, respectively). Does it mean that 1.7 (6.5-4.8) NUMTs are somatically acquired or cancer-specific? Then, because the authors analyzed 12,509 cancer genomes, they should have found ~20,000 cancer-specific NUMTs (or de novo; $12,509 \times 1.7$). However, they report that they found only 379 de novo NUMTs (which rate is in line with previous reports, Yuan et al.,).

Therefore, my interpretation of the 6.5 NUMTs is that not all of them are somatically acquired and thus should be of germline origin. In this scenario, tumor genomes should not harbor more NUMTs. Instead, they should show fewer NUMTs than matched normal tissues as many chromosomes are lost (in the form of loss of heterozygosity) in cancer. The authors should clarify why they found more NUMTs in the cancer genome.

(5-new question) In the "Molecular evolution of NUMT sequences" section, the authors introduce point mutation in the NUMT segments. It is good. However, I do not clearly understand why the authors categorize these variants into three groups - Groups A, B and C. More clarification on the meaning of grouping is appreciated.

Referee #3 (Remarks to the Author):

I found the Authors responsive to critiques and comments. The response was carefully done and thoughtfully crafted. My concerns are satisfied.

Author Rebuttals to First Revision:

Nature 2022-01-00173A: response to referees' comments

Referee #1:

Referee: A minor note: I did try to access their NUMT database, but could not do so as it was password protected. I also was unable to find a pointer to the code associated with analyzing the data, which would likely be useful for the field.

Response: The password for the NUMT database (<https://wwi.shinyapps.io/numts/>) was included in the **Web Resources** section of the revised manuscript (P16):

User name: reviewer

Password: reviewer_numts

We will remove the password protection on publication.

The code used for our analysis was indicated in the **Code availability** section of the revised manuscript (P16) and is available on GitHub through the following URL:

<https://github.com/WeiWei060512/NUMTs-detection.git>

Referee #2:

Referee: In the revised manuscript entitled "Characterization of nuclear-embedded mitochondrial DNA sequences in 66,083 human genomes", Wei and colleagues reports novel NUMTs in human germline and cancer cells. As Reviewer #1 also pointed out, it is not easy to find a conceptual innovation. However, the beauty of this study comes from its huge sample size (>60K whole-genomes). It provides the most comprehensive analysis so far, and thus it will be an excellent resource for future studies. Therefore, together with outcomes from the downstream analyses, the list of NUMT calls is one of this manuscript's essential components.

Although the authors revised the manuscript according to my suggestion, I still feel some parts are not very clear. Therefore, I request some more clarification before its publication in

the journal.

(1-in line with my previous question) The NUMT calls are now listed in Extended Data Table 1. I appreciate the effort. However, some more information may be necessary. For example, only NUMT frequency is categorized by a few groups, such as common, rare, and ultra-rare. The actual frequency can be shown in the table - potentially in each ethnic group. Ideally, showing sample id harboring specific NUMTs will be helpful for future studies. If the data size is not too big (I believe so), this information should be shared through an extended data table, not by the authors' website as it would not be as stable as the archive of the journal. In addition, the authors' website (<https://wwwei.shinyapps.io/numts/>) can be accessed after the email address is registered.

Response: We have added six additional columns to Extended Data Table 1. These show the total frequency of each NUMT and the frequency subdivided into five ethnic groups. We will remove the password protection of the NUMT website (<https://wwwei.shinyapps.io/numts/>) on publication.

Referee: (2-in line with my previous question) Throughout this manuscript, the definition of 'identified NUMT' is confusing. The human reference genome sequence, mostly contributed by a European individual, already has ~700 NUMTs ("referenced NUMTs"; Simone D et al., BMC Genomics 12:517), the vast majority of which, but not all, should be universal (shared by all human individuals) or common (shared by many individuals) NUMTs. Therefore, most human individuals have hundreds NUMTs in the genome (many of them should be 'referenced'). In this sense, the NUMT number reported by the authors (4.7 NUMTs; page 3; Fig 1d) may cause a lot of confusion. My interpretation of the 1,615 NUMTs reported here is 'the NUMTs absent in the human reference genome' ("non-referenced NUMTs"). Otherwise, 4.7 should be too small. It should be explicitly described in the first section of the Results.

Response: The reviewer is correct. Our study was designed to characterise NUMTs not present in the human reference genome. We welcome the opportunity to make this clearer in the revised manuscript.

P5 Methods – title of the relevant section

Detecting NUMTs and breakpoints not present in the reference sequence

P18 Legend to Fig. 1d

Histogram of the average number of NUMTs per individual that were not present in the reference sequence and were detected by at least 5 pairs of discordant reads.

P24 Legend to Fig. 4a

Histogram of the average number NUMTs detected per normal (in navy colour) and tumour (in red colour) samples that were not present in the reference sequence.

We have also updated, the x-axis label on Fig. 1d and Fig. 4a to make this totally explicit:

Fig. 1d and Fig. 4a

Number of NUMTs not in reference sequence

Referee: (2-a) The meaning of "non-referenced NUMTs" is not identical to "novel NUMTs", because some of the 1,615 non-referenced NUMTs should be already reported by previous papers (e.g., Dayama et al., Nucleic Acids Research).

Response: The reviewer is correct. We detected 1,637 NUMTs not present in the reference sequence. 1,615 if these NUMTs were defined as rare or ultra-rare. 1,564 of these NUMTs were not reported in any previous publications (Extended Data Table 2)¹⁻⁴. We have made this clearer in the Atlas section as follows (note, this is the original text with the addition of 'nor present in the reference sequence' to make it clear what the numbers are referring to). We also include reference to a new Extended Data Table as requested by the reviewer's next point.

P3/4

Individuals had an average of 4.7 NUMTs (s.d = 1.6) not present in the reference sequence (Fig. 1d). There was no difference between the males/females (P value = 0.834, Wilcoxon rank sum test, Extended Data Fig. 3a) nor with age (P value = 0.95, Pearson's correlation) (Extended Data Fig. 3b). 1615 different NUMTs (98.7%) not present in the reference sequence seen in 26.2% individuals were rare/ultra-rare (F<1%), 1567 different NUMTs (96.1%) seen in 14.2% individuals were ultra-rare (F<0.1%), and 1039 (63.7%) NUMTs were seen only one family (3.6%) (Fig. 1e, Extended Data Fig. 2a). As expected, the majority (71.4%) of the common NUMTs (F >=1%) were previously reported (Extended Data Table 2)¹⁻⁴. Thus, combining the rare/ultra-rare and common NUMT data, we identified 1564 novel NUMTs not previously reported (Fig. 1e, Extended Data Table 1).

Referee: (2-b) If referenced NUMTs are not included in the list of 1,615 NUMTs as I supposed, there should be another concern. Not all the referenced NUMTs are shared by all human individuals. In extreme cases, some referenced NUMTs should be ultra-rare or private to the reference genome. Some rare NUMTs may not be counted in this manuscript, if it is referenced. If this scenario is true, Fig 1d should not be very meaningful, because it must be affected by the reference genome sequences. For example, the individual whose genome was used for the human genome project, will have no NUMT in the genome in this scenario. The NUMT number per genome should be counted absolutely.

(2-c) Therefore, I believe that referenced but polymorphic NUMTs should also be included (or considered) in this manuscript if the authors wish to count the number of NUMTs per individual, as shown in Fig 1d. Otherwise, the authors should at least explicitly describe that what they count here is 'non-referenced NUMTs'. In the latter case, the list of NUMTs reported in the manuscript is biased, and the impact of this study as an encyclopedia of NUMTs may be a bit compromised.

Response to (2-b) and (2-c): We have now included Extended Data Table 2 which lists the NUMTs present in the reference sequence and previously published NUMTs¹⁻⁴. This emphasises that the results we present in this manuscript relate to NUMTs not present in the reference sequence, and also allows the reader to easily access data on all known NUMTs, including those present in the reference genome, those previously reported, and those described in our manuscript for the first time.

Referee: (2-d) It is helpful if the authors can also show their concept (i.e., referenced or non-referenced) of NUMT in Fig 1a.

Response: We have altered Fig 1a to make it explicit that we focus on NUMTs not present in the reference sequence. We have also expanded the figure legend to make this clear:

P17 Fig. 1a: Text of 2nd step

Discover NUMTs not in ref seq

P18 Legend to Fig. 1a

Bioinformatic pipeline for the detection of NUMTs not present in the reference sequence.

Referee: (3-in line with my previous question) The length distribution of NUMTs ranges from 24bp to the whole mt genome. My previous question was how we could make sure that the origin of the three de novo NUMT is mitochondria in the germline of the parents. There are hundreds of (referenced) NUMTs in the genome, and those may be the source of the insertions. If the inserted sequences have mtDNA polymorphisms found in parents' germline DNA, it could be supporting evidence. Unfortunately, the authors did not find such variants. Although the authors claim that the NUMT sequences did not align to any other site in the nuclear genome of the child (reference genome sequence?-how can we align reads in the nuclear genome of the child?) it could not be perfect supporting evidence. I am wondering about the mapping quality of the NUMT supporting reads. The origin of the NUMT sequences could be the parents' non-referenced NUMTs.

Response: As described in the previous revision, we performed *de novo* genome assembly for each child harbouring a *de novo* NUMT. We found no additional matches for the *de novo* NUMT sequence in the child's genome, so it is highly unlikely that the *de novo* NUMT originated from the nuclear DNA. We also excluded the possibility that the *de novo* NUMT

originated from the reference sequence. The *de novo* NUMTs were also not present in the published NUMT sequence databases (now included as Extended Data Table 2)

To provide further reassurance about the mapping strategy, we increased the sensitivity of for NUMT detection in this analysis by dropping the requirement for ≥ 5 discordant read pairs to two discordant read pairs in each child and their parents. We still found no additional match for the *de novo* NUMT. This gives us additional confidence that these specific NUMTs were indeed *de novo*.

P5

In each case the de novo NUMT sequence did not align to any other site in assemblies of the nuclear genomes of the child, making it unlikely that the NUMTs originated from within the nuclear DNA. None of other NUMTs detected in each child and their parents carried the same NUMT sequence as the de novo NUMT insertions, even after increasing the mapping sensitivity by dropping the requirements from ≥ 5 discordant reads to two discordant reads. The de novo NUMTs were also not present in the reference genome, nor in published NUMT lists (Extended Data Tables 2).

Referee: (3-a) Although the authors introduced a new sentence in the revised manuscript that claims "the *de novo* NUMT frequency is likely to be an under-estimate...", I believe that the rate can be "over-estimate" as the origin of the insertion could be 'nuclear DNA'.

Response: We accept the reviewer's point and have altered the text as follows:

P5

The de novo NUMT frequency is likely to be an under-estimate because of the difficulty determining the origin of short NUMTs, although we cannot absolutely exclude the possibility of apparent de novo NUMTs arising from other parts of the nuclear genome and not from a new mtDNA insertion event.

Referee: (4-in line with my previous question) I still do not understand why the authors found two more NUMTs in the cancer genome (6.5 v.s. 4.8 in cancers and matched normals, respectively). Does it mean that 1.7 (6.5-4.8) NUMTs are somatically acquired or cancer-specific? Then, because the authors analyzed 12,509 cancer genomes, they should have found ~20,000 cancer-specific NUMTs (or *de novo*; $12,509 \times 1.7$). However, they report that they found only 379 *de novo* NUMTs (which rate is in line with previous reports, Yuan et al.). Therefore, my interpretation of the 6.5 NUMTs is that not all of them are somatically acquired and thus should be of germline origin. In this scenario, tumor genomes should not harbor more NUMTs. Instead, they should show fewer NUMTs than matched normal tissues

as many chromosomes are lost (in the form of loss of heterozygosity) in cancer. The authors should clarify why they found more NUMTs in the cancer genome.

Response: The apparent discrepancy relates to our stringent filtering approach employed before we analysed the characteristics of the *de novo* NUMTs in tumours. The reviewer is correct that cancers had ~1.7 more NUMTs than matched normal tissue on average. However, to be absolutely certain that a NUMT was somatically acquired or cancer specific, we performed an additional filtering step by removing any NUMTs present in any other non-cancer sample in our dataset. In this way we were totally confident that the remaining 379 NUMTs were, indeed, cancer specific.

We have made this clearer in the revised manuscript as follows:

P7

Next, we focussed on a subgroup of the tumour-specific NUMTs not present in any other non-cancer genome, giving us high confidence that these NUMTs arose either in somatic tissues leading to the cancer, or the cancer itself. 379 of these de novo NUMTs were seen in 251 tumours (2.3%) from 10,713 tumour-normal pairs, giving a rate of 3.56×10^{-2} per cancer per genome

Referee: (5-new question) In the "Molecular evolution of NUMT sequences" section, the authors introduce point mutation in the NUMT segments. It is good. However, I do not clearly understand why the authors categorize these variants into three groups - Groups A, B and C. More clarification on the meaning of grouping is appreciated.

Response: We studied the three different groups to determine whether more stringent filtering influenced our interpretation of the results. We have made this clear in the revised manuscript as follows:

P10

We studied NUMT-specific variants in three categories using increasingly stringent filtering criteria to determine whether the filtering had a major impact on our interpretation. The categories were as follows:.....

Referee #3:

Referee: I found the Authors responsive to critiques and comments. The response was carefully done and thoughtfully crafted. My concerns are satisfied.

Response: We thank the reviewer for their help in improving our manuscript.

References in the response to reviewers

1. Simone, D., Calabrese, F.M., Lang, M., Gasparre, G. & Attimonelli, M. The reference human nuclear mitochondrial sequences compilation validated and implemented on the UCSC genome browser. *BMC Genomics* **12**, 517 (2011).
2. Calabrese, F.M., Simone, D. & Attimonelli, M. Primates and mouse NumtS in the UCSC Genome Browser. *BMC Bioinformatics* **13 Suppl 4**, S15 (2012).
3. Li, M., Schroeder, R., Ko, A. & Stoneking, M. Fidelity of capture-enrichment for mtDNA genome sequencing: influence of NUMTs. *Nucleic Acids Res* **40**, e137 (2012).
4. Dayama, G., Emery, S.B., Kidd, J.M. & Mills, R.E. The genomic landscape of polymorphic human nuclear mitochondrial insertions. *Nucleic Acids Res* **42**, 12640-12649 (2014).

Reviewer Reports on the Second Revision:

Referees' comments:

Referee #1 (Remarks to the Author):

I thank the authors for their responses to my and other comments, and commend them for a thorough and rigorous response to all the referee feedback.

Regarding my two queries, I have now accessed both the github repository and the NUMT database. The github repository includes scripts and other files that are associated with the identification of NUMTs and related analyses. However, there is no documentation at all supporting this code, e.g. indicating in detail the necessary configuration, dependencies, and other parameters required for one to run this code on their own BAM files. It's unclear how each script is related to key data or analyses in the manuscript. This point seems particularly critical as others may be eager to now build on this work by replicating the analyses on their own datasets. Finally, as there are some novel statistical analyses in the manuscript, I would recommend to the authors that they include the .R or .py scripts required to generate the results/figures associated with these analyses (e.g. by reading in key underlying data and then applying a statistical test) in this github repository as well.

The NUMT database is similarly undocumented. Some of the fields and the entries in the fields are ambiguous (e.g. "UN" in concatenatedNUMTs). This is remedied readily by simply providing a thorough set of documentation describing each field and its particular values in a separate tab on the shiny app. This will make the database far more useful. Is the data in this database fully available for bulk download?

Referee #2 (Remarks to the Author):

No more concerns for this manuscript. Congratulations.

Author Rebuttals to Second Revision:

Nature 2022-01-00173B

Response to Referee #1

I thank the authors for their responses to my and other comments, and commend them for a thorough and rigorous response to all the referee feedback.

Regarding my two queries, I have now accessed both the github repository and the NUMT database. The github repository includes scripts and other files that are associated with the identification of NUMTS and related analyses. However, there is no documentation at all supporting this code, e.g. indicating in detail the necessary configuration, dependencies, and other parameters required for one to run this code on their own BAM files. It's unclear how each script is related to key data or analyses in the manuscript. This point seems particularly critical as others may be eager to now build on this work by replicating the analyses on their own datasets. Finally, as there are some novel statistical analyses in the manuscript, I would recommend to the authors that they include the .R or .py scripts required to generate the results/figures associated with these analyses (e.g. by reading in key underlying data and then applying a statistical test) in this github repository as well.

The NUMT database is similarly undocumented. Some of the fields and the entries in the fields are ambiguous (e.g. "UN" in concatenatedNUMTs). This is remedied readily by simply providing a thorough set of documentation describing each field and its particular values in a separate tab on the shiny app. This will make the database far more useful. Is the data in this database fully available for bulk download?

We are pleased to provide the relevant documentation and annotation which is now included on the github repository and the welcome page of the NUMT database. We have made the database fully downloadable.